# An alternative mechanism by which If1 prevents ATP hydrolysis by the ATP synthase subcomplex in *S. cerevisiae*

Orane Lerouley [ID][1], Isabelle Larrieu[1,5], Tom Louis Ducrocq [ID][1], Benoît Pinson [ID][1,2], Marie-France Giraud [ID][3] & Arnaud Mourier [ID][1,4][✉]

## Abstract

The mitochondrial $F_1F_0$-ATP synthase is crucial for maintaining the ATP/ADP balance which is critical for cell metabolism, ion homeostasis and cell proliferation. This enzyme, conserved across evolution, is found in the mitochondria or chloroplasts of eukaryotic cells and the plasma membrane of bacteria. In vitro studies have shown that the mitochondrial $F_1F_0$-ATP synthase is reversible, capable of hydrolyzing instead of synthesizing ATP. In vivo, its reversibility is inhibited by the endogenous peptide If1 (Inhibitory Factor 1), which specifically prevents ATP hydrolysis in a pH-dependent manner. Despite its presumed importance, the loss of If1 in various model organisms does not cause severe phenotypes, suggesting its role may be confined to specific stress or metabolic conditions yet to be discovered. Our analyses indicate that inhibitory peptides are crucial in mitigating mitochondrial depolarizing stress under glyco-oxidative metabolic conditions. Additionally, we found that the absence of If1 destabilizes the nuclear-encoded free $F_1$ subcomplex. This mechanism highlights the role of If1 in preventing harmful ATP wastage, offering new insights into its function under physiological and pathological conditions.

Keywords IF1; ATP Synthase; F1 Subcomplex; Mitochondria; Bioenergetics
Subject Categories Membranes & Trafficking; Metabolism

## Introduction

The mitochondrial $F_1F_0$-ATP synthase is among the most advanced molecular enzymatic nanomachines in the living word. This enzyme is highly evolutionary conserved (Sinha and Wideman, 2023), and ubiquitous in the mitochondria or chloroplasts of eukaryotic cells, as well as in the plasma membrane of bacteria (Lau et al, 2008; Hahn et al, 2018; Gu et al, 2019; Pinke et al, 2020; Yang et al, 2020; Courbon and Rubinstein, 2022). This multisubunit enzyme is a cornerstone of the oxidative phosphorylation system (OXPHOS) as it transduces the proton electrochemical gradient ($\Delta\mu H^+$) generated by the respiratory chain, to synthesize ATP from ADP and inorganic phosphate (Mitchell, 1961; Boyer et al, 1973; Stock et al, 1999; Watt et al, 2010). In *Saccharomyces cerevisiae*, this 600 kDa enzyme comprises a catalytic domain $F_1$ ($\alpha_3\beta_3\gamma_1\delta_1\varepsilon_1$) and a $F_0$ region, divided into a membranous rotor (subunit (su) $9_{10}$-ring) and a peripheral stalk (su 4, su 6, su 8, su f, OSCP, su d, su h, su i/j) that connects the catalytic head to the rotor ring. Three additional subunits, su e, su g, and su k are involved in enzyme dimerization. Proton translocation across two hemi-channels at the interface of su 6 and the membranous ring induces ring rotation, that triggers the rotation of a central stalk ($\gamma,\delta,\varepsilon$) inside the two catalytic subunits ($\alpha$ and $\beta$), enabling conformational changes required for ATP synthesis. The eukaryotic $F_1F_0$-ATP synthases are composed of 17 different subunits, encoded by the nuclear (nDNA) or mitochondrial genome (mtDNA) (Senior, 1988; Kühlbrandt, 2019). The dual genetic origin of the $F_1F_0$-ATP synthase implies that gene expression from both genomes must be tightly coordinated to ensure proper biogenesis and assembly of the enzyme. Nevertheless, in cells presenting defective mitochondrial genome levels and expression or impaired ATP synthase assembly, $F_1$ is commonly found assembled as a stable subcomplex capable of ATP hydrolysis (Tzagoloff, 1969; Carrozzo et al, 2006; Wittig et al, 2010). The $F_1F_0$-ATP synthase, along with glycolysis and other pathways that allow substrate-level phosphorylation, is critical in maintaining the ATP/ADP balance, which is required for cell metabolism, ion homeostasis, cell division, proliferation, and motility. In multicellular organisms, mitochondrial ATP synthesis is finely adjusted to sustain specialized functions of differentiated cells, and in humans, defective OXPHOS-driven ATP synthesis causes multiple and severe diseases frequently affecting high-energy demanding tissues such as cardiac and skeletal muscles, as well as the nervous system (Galber et al, 2021).

Interestingly, the mitochondrial $F_1F_0$-ATP synthase is fully reversible and in vitro experiments performed on purified enzymes or functional mitochondria demonstrated that ATP hydrolysis could be coupled to proton translocation, generating a proton electrochemical potential across the inner mitochondrial membrane (Boyer

[1]University of Bordeaux, CNRS, IBGC, UMR 5095, 33000 Bordeaux, France. [2]Metabolic Analyses Service, TBMCore-Université de Bordeaux-CNRS UAR 3427-INSERM, US005 Bordeaux, France. [3]University of Bordeaux, CNRS, Bordeaux INP, CBMN, UMR 5248, F-33600 Pessac, France. [4]Division of Molecular Metabolism, Department of Medical Biochemistry and Biophysics, Karolinska Institutet, Stockholm, Sweden. [5]Deceased: Isabelle Larrieu. [✉]E-mail: arnaud.mourier@ibgc.cnrs.fr

et al, 1973; Pietrobon et al, 1983; Mourier et al, 2010). However, the reversibility of the $F_1F_0$-ATP synthase is, under physiological conditions, prevented by the membrane potential generated by the respiratory chain. This reversed activity is only observed when the respiratory chain is blocked (chemical inhibitor or anoxia), or when the proton electrochemical membrane potential is abolished. The reversibility of the $F_1F_0$-ATP synthase is also regulated by a nuclear encoded inhibitory peptide, so called inhibitory factor 1 (If1), which can physically interact and inhibit the $F_1$ catalytic domain (Pullman and Monroy, 1963). Since its discovery, homologs of If1 were found and characterized in other species (Cintrón and Pedersen, 1979; Hashimoto et al, 1981; Matsubara et al, 1981; Norling et al, 1990; Ichikawa and Ogura, 2003). The If1 amino acid sequences are well conserved across evolution, and for yet unclear reasons, two homologous inhibitory peptides, namely If1 and Stf1 (Stabilizing Factor 1), presenting redundant activity were identified in S. cerevisiae (Hashimoto et al, 1987; Cabezon et al, 2002; Venard et al, 2003). If1 and Stf1 proteins are encoded respectively by INH1 and STF1 nuclear gene in S. cerevisiae. An important wealth of experiments on independent eukaryote models have contributed to characterize the mechanism of action (Pullman and Monroy, 1963; Hashimoto et al, 1981, 1984; Cabezon et al, 2000, 2002; Venard et al, 2003), as well as the structural interaction of these peptide inhibitors with the $F_1F_0$-ATP synthase (Cabezón et al, 2001; Robinson et al, 2013; Boreikaite et al, 2019; Gu et al, 2019; Mühleip et al, 2021; Romero-Carramiñana et al, 2023). One of the most remarkable and evolutionary conserved features of If1 inhibition is its regulation by pH, being optimal under neutral or slightly acidic pH conditions and inactive at pH above 8.0 (Pullman and Monroy, 1963; Hashimoto et al, 1987). The pH-dependent If1 inhibition of $F_1F_0$-ATP synthase strikingly aligns and supports its function, potentiating its capacity to prevent ATP hydrolysis under depolarization when the ΔpH is abolished. Interestingly, the importance of If1 under genetic or chemical stress preventing maintenance of the membrane potential by the respiratory chain has been confirmed in various model organisms (Buchet and Godinot, 1998; Rouslin and Broge, 1996; Sgarbi et al, 2018; Venard et al, 2003).

The Cryo-EM structures of oligomeric $F_1F_0$-ATP synthases demonstrated that If1 dimers, could bridge adjacent $F_1F_0$-ATP synthase dimers, suggesting that If1 could stabilize oligomers (Cabezón et al, 2000; Pinke et al, 2020; Gu et al, 2019). Functional investigations in mammalian cells and mouse models supported the idea that If1 regulates $F_1F_0$-ATP synthase oligomerization (Domínguez-Zorita et al, 2023) and even suggested that If1 could also regulate ATP synthesis (García-Bermúdez et al, 2015; Sánchez-Cenizo et al, 2010). However, the role of If1 in controlling $F_1F_0$-ATP synthase oligomerization and ATP synthesis activity remains debated and needs to be confirmed in other model organisms (Dienhart et al, 2002; Lucero et al, 2021; Gatto et al, 2022; Carroll et al, 2024; Galkina et al, 2022). The current controversy over the role of If1 in energy metabolism partly arises from the lack of methods to monitor, in vivo, ATP hydrolysis by the mitochondrial ATP synthase operating in reverse. Furthermore, the absence of major phenotypes associated with If1 loss in many organisms suggests that its action may be limited to specific stress or metabolic conditions that remain to be discovered (Ichikawa et al, 1990; Nakamura et al, 2013; Fernández-Cárdenas et al, 2017).

The goal of our study was to clarify the structural and physiological roles of inhibitory peptide If1/Stf1 in the yeast S. cerevisiae. Our analyses demonstrate that the If1/Stf1 activity is dispensable to sustain the growth of yeast under 'respiro-fermentative' or 'respiratory strict' carbon sources. However, we observed that inhibitory peptides are key in sustaining growth of yeast subjected to mitochondrial depolarizing stress under glyco-oxidative metabolic conditions. We also hereby demonstrate that loss of inhibitory peptides does not impact high supramolecular organization of the yeast $F_1F_0$-ATP synthase but surprisingly destabilizes the free $F_1$ subcomplex. This discovery prompted us to revisit the role of the free $F_1$ subcomplex to sustain the ability for S. cerevisiae to grow in total or partial absence of mitochondrial genome ($\rho^{-/o}$).

## Results

### If1/Stf1 inhibitors are required to maintain the ATP synthase-free $F_1$ subcomplex

Independent works have previously established that S. cerevisiae expresses two $F_1F_0$-ATP synthase inhibitory peptides named If1 and Stf1, respectively encoded by the genes INH1 and STF1 (Ichikawa et al, 1990; Hashimoto et al, 1990). Therefore, to investigate the role of $F_1F_0$-ATP synthase endogenous inhibitory peptides on yeast energy producing system and metabolism, we generated an inh1Δ stf1Δ double knockout strain. The complete loss of If1 and Stf1 was validated by Western blot analyses performed on total protein extracts from yeast harvested during exponential growth on non-fermentable carbon source (glycerol 2%) (Fig. 1A). We observed that the individual or combined loss of the peptide inhibitors did not affect the growth on respiratory strict carbon sources such as lactate (2%) (Fig. 1B). This unaltered growth on lactate carbon source, which depends on mitochondrial OXPHOS content and activity (Devin et al, 2006), suggested that the loss of If1 and Stf1 did not strongly affect OXPHOS capacities under physiological conditions. We then performed classical native polyacrylamide gel electrophoresis (PAGE) to characterize the supramolecular assembly of the $F_1F_0$-ATP synthase in inh1Δ stf1Δ purified mitochondria (Fig. 1C). The in-gel ATPase activity demonstrated that, in line with previous reports (Dienhart et al, 2002), the $F_1F_0$-ATP synthase monomers (V) and dimers were unchanged in inh1Δ stf1Δ. Interestingly, our native PAGE experiments demonstrate that levels of higher $F_1F_0$-ATP synthase oligomers were not impacted by the combined loss of both If1 and Stf1 (Fig. 1C). However, in contrast to the $F_1F_0$-ATP synthase oligomers, we noticed that the free $F_1$ subcomplex level was almost undetectable in inh1Δ stf1Δ. The free $F_1$ subcomplex used to be frequently interpreted as a degradation or destabilization byproduct of the $F_1F_0$-ATP synthase monomers or dimers, potentially occurring during mitochondrial isolation or detergent solubilization. To minimize the risk of degradation, we decided to characterize the $F_1F_0$-ATP synthase supramolecular organization on total soluble protein extract bypassing potential mitochondrial degradation inherent to the fastidious mitochondrial isolation procedure. Despite dampening the resolution and characterization of high molecular weight complexes, blue native PAGE (BN-PAGE) performed on total cell extracts confirmed that free $F_1$ subcomplex

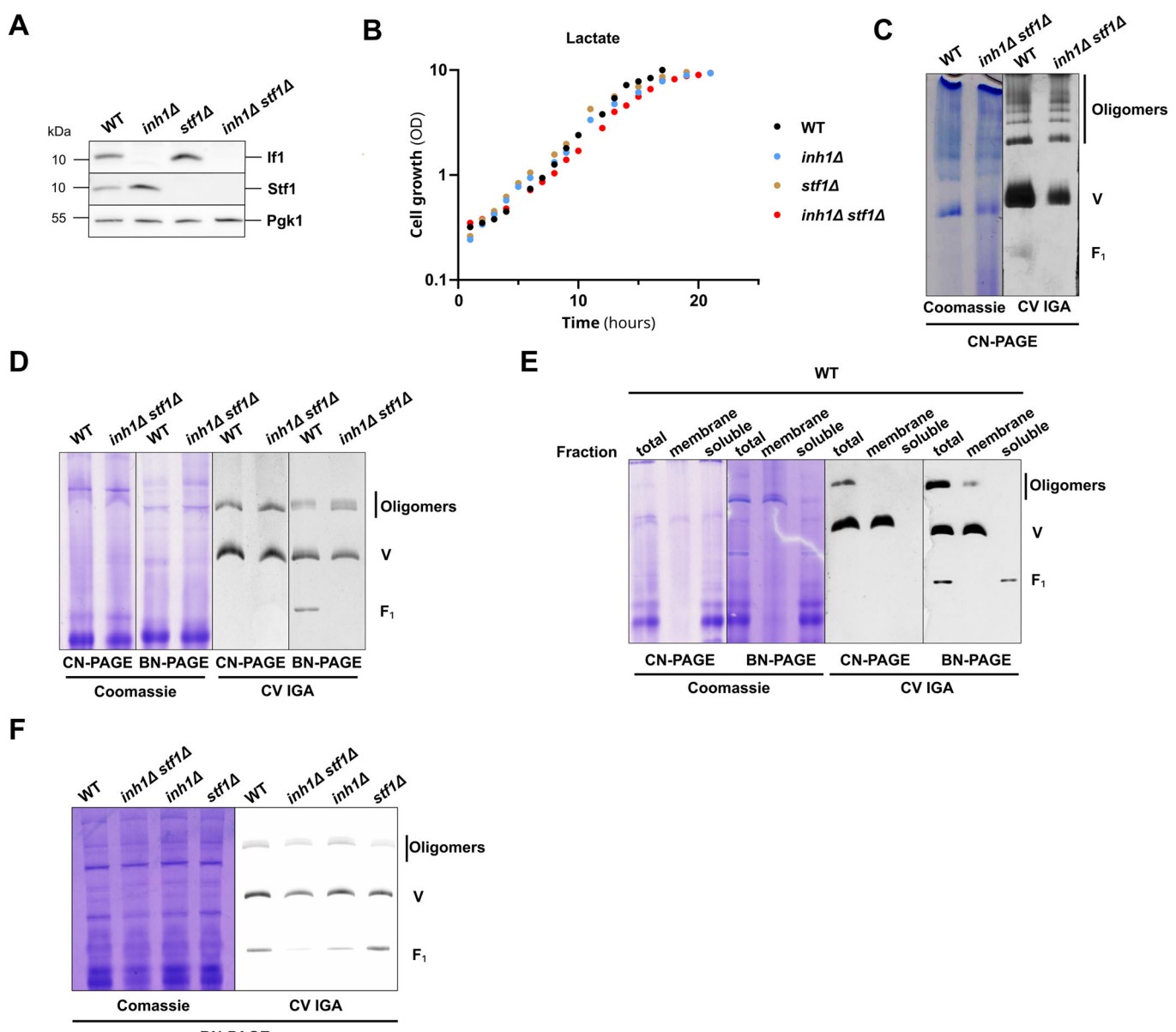

**Figure 1.   If1/Stf1 are required to maintain the $F_1F_0$-ATP synthase-free $F_1$ subcomplex levels.**

(A) Western blot performed on total cell protein extracts purified from WT, *inh1Δ*, *stf1Δ*, and *inh1Δ stf1Δ* mutants, grown on glycerol 2% rich medium. (Representative of *n* = 5 independent experiments). (B) Growth of WT (black), *inh1Δ* (blue), *stf1Δ* (brown), and *inh1Δ stf1Δ* (red) mutants on lactate 2% rich medium, following the optical density of the culture at 550 nm. (*n* = 5 independent experiments). (C) CN-PAGE (3–12%) performed with purified mitochondria from WT and *inh1Δ stf1Δ* cells grown on lactate 2% rich medium, solubilized with glyco-diosgenin (GDN) at a GDN to protein ratio of 0.5 g/g protein. The $F_1F_0$-ATP synthase assemblies were revealed by $F_1F_0$-ATP synthase hydrolytic in-gel activity (CV IGA). (Representative of *n* = 3 independent experiments). (D) CN and BN-PAGE (3–12%) performed with total cell extracts from WT and *inh1Δ stf1Δ* grown on glycerol 2% rich medium solubilized with digitonin at a digitonin-to-protein ratio of 1.5 g/g protein. The $F_1F_0$-ATP synthase assemblies were revealed by $F_1F_0$-ATP synthase hydrolytic in-gel activity (CV IGA). An extended version of both Coomassie and IGA staining are presented in Fig. 3C (Representative of *n* = 3 independent experiments). (E) CN and BN-PAGE (3–12%) performed with total cell extracts, membrane and soluble fractions obtained after ultracentrifugation of WT cells grown on glycerol 2% rich medium solubilized with digitonin at a digitonin-to-protein ratio of 1.5 g/g protein. The $F_1F_0$-ATP synthase assemblies were revealed by $F_1F_0$-ATP synthase hydrolytic in-gel activity (CV IGA). (Representative of *n* = 3 independent experiments). (F) BN-PAGE (3–12%) performed with total cell extracts from WT, *inh1Δ stf1Δ*, *inh1Δ*, and *stf1Δ*, grown on glycerol 2% medium solubilized with digitonin at a digitonin-to-protein ratio of 1.5 g/g protein. The $F_1F_0$-ATP synthase assemblies were revealed by $F_1F_0$-ATP synthase hydrolytic in-gel activity (CV IGA). (Representative of *n* = 3 independent experiments). Source data are available online for this figure.

was present in control yeast but lost in *inh1Δ stf1Δ* (Fig. 1D). Intriguingly, in contrast to the $F_1F_0$-ATP synthase monomers and oligomers levels, the free $F_1$ subcomplex was clearly detected in WT under BN-PAGE and hardly visible under clear native PAGE (CN-PAGE) conditions. This observation prompted us to determine if

the free $F_1$ subcomplex observed in BN-PAGE could result from (i) the potential impact of the Coomassie brilliant blue on destabilization of the fully assembled complexes (V and oligomers) or (ii) the fact that migration of the free $F_1$ relies on its binding to the charged Coomassie dye. To this end we performed CN or BN-PAGE to

characterize the $F_1F_0$-ATP synthase assemblies present in digitonin solubilized proteins in (i) total cell extracts, (ii) total cell membrane extracts, and (iii) total cell soluble fraction (Fig. 1E). As expected, the membrane and soluble fractionation could efficiently separate the membrane-anchored fully assembled ATP synthase from the membrane-free $F_1$ subcomplex, confirming that the free $F_1$ subcomplex is a soluble entity. Moreover, the absence of the free $F_1$ subcomplex in the solubilized membrane extracts demonstrated that this subcomplex is not a destabilization byproduct of the fully assembled ATP synthase post solubilization. Our conclusion, supporting that free $F_1$ subcomplex is not a destabilization byproduct was strengthened by the titration of the ratio between digitonin and mitochondrial protein (Fig. EV1A,B). We observed that the progressive increase in digitonin-to-protein ratio gradually destabilized oligomers, but did not impact the levels of free $F_1$ subcomplex detected in WT or in $inh1\Delta$ $stf1\Delta$. Consequently, the faint level of free $F_1$ observed in all extracts subjected to CN-PAGE suggested that the migration of the soluble free $F_1$ subcomplex is heavily conditioned by the Coomassie-conferred charge (Fig. 1E). In line with the result obtained previously (Fig. 1D,E), BN-PAGE analysis of total protein showed that the $F_1$ subcomplex level was more severely reduced in $inh1\Delta$ than in $stf1\Delta$, and was hardly detected in $inh1\Delta$ $stf1\Delta$ (Fig. 1F). Altogether, our analyses clearly indicate that the amounts of free soluble $F_1$ subcomplex rely on If1 and to a lower extend on Stf1.

## If1 binds and inhibits ATP synthase oligomers, monomers and free $F_1$ subcomplex

The intriguing interdependency between If1/Stf1 and free $F_1$ subcomplex prompted us to further investigate the interplay between these factors. First, to gain structural insights into the interaction between If1 and the mitochondrial $F_1F_0$-ATP synthase, we performed two-dimensional electrophoresis i.e., BN-PAGE followed by a second dimensional gel, denaturing SDS-PAGE (Fig. 2A). The Western blot experiments confirmed that the entities identified so far through their in-gel ATPase activities (Fig. 1C–E), were indeed the mitochondrial $F_1F_0$-ATP synthase complexes and subcomplexes (Fig. 2A) and confirmed the drastic loss of free $F_1$ in $inh1\Delta$ $stf1\Delta$. Beyond some specific $F_1F_0$-ATP synthase subunits, we also managed to localize If1 proteins and confirmed that If1 physically interacts with the different $F_1F_0$-ATP synthase assemblies (Fig. 2A,B). Interestingly, the densitometric signal quantification demonstrated that If1 exhibits an even binding capacity toward the different $F_1F_0$-ATP synthase assemblies (Fig. 2B). However, we could not detect or visualize Stf1 in the second dimension. The previously reported lower binding efficiency and affinity of Stf1 for $F_1F_0$-ATP synthase compared to If1 (Venard et al, 2003), could likely explain the undetectable level of Stf1 following the digitonin extraction and BN-PAGE procedures. Next, we functionally characterized the interplay between If1/Stf1 and $F_1F_0$-ATP synthase assemblies on non-solubilized samples (Fig. 2C–E). The ATP hydrolysis flux measurement performed on total yeast protein extracts confirmed that the pH-dependent inhibition of the ATPase activity was completely abolished in $inh1\Delta$ $stf1\Delta$ (Fig. 2C). Furthermore, our analyses demonstrated that the oligomycin-sensitive ATPase activity, associated with fully assembled $F_1F_0$-ATP synthase, assessed in WT and $inh1\Delta$ $stf1\Delta$ samples were identical (Fig. 2D). In contrast, the oligomycin-insensitive ATPase activity,

mainly related to free $F_1$ subcomplex, was drastically reduced in the $inh1\Delta$ $stf1\Delta$ strain (Fig. 2D). The complete oligomycin insensitivity of the ATP hydrolysis activity assessed in the soluble fraction (Fig. 2E), containing exclusively free $F_1$ subcomplexes (Fig. 1E), confirmed that the oligomycin-resistance was inherent to free $F_1$ subcomplexes. Interestingly, the oligomycin-resistant ATPase activity of free $F_1$ subcomplexes was fully inhibited by If1/Stf1 through their characteristic pH-dependent inhibition (Fig. 2E), confirming their capacity to physically and functionally interact (Fig. 2A,B). This functional characterization nicely corroborates the structural observation showing that WT and $inh1\Delta$ $stf1\Delta$ present similar levels of $F_1F_0$-ATP synthase monomers and oligomers (Fig. 1C,D). Altogether, native PAGE (Fig. 1C–E) and functional analyses (Fig. 2D,E) demonstrate that the oligomycin-insensitive free $F_1$ subcomplex is severely reduce in $inh1\Delta$ $stf1\Delta$.

## If1/Stf1 are specifically involved in free $F_1$ subcomplex stabilization

To further investigate the interplay between the inhibitory peptides If1/Stf1 and the free $F_1$ subcomplex, we decided to evaluate the stability of the different $F_1F_0$-ATP synthase assemblies in the $inh1\Delta$ $stf1\Delta$ strain. To this end, we characterized the fate of the different ATP synthase assemblies in WT and $inh1\Delta$ $stf1\Delta$ strains grown in complete medium containing glycerol (2%) and subjected to a cycloheximide treatment inhibiting cytosolic translation (Buchanan et al, 2016). This experiment clearly showed that a 90-min treatment did not affect the steady state levels of the inhibitory peptides If1/Stf1 but severely affected, with distinct kinetics, the levels of $F_1F_0$-ATP synthase assemblies (Fig. 3A,B). Interestingly, the destabilization profile of $F_1F_0$-ATP synthase monomers and oligomers in $inh1\Delta$ $stf1\Delta$ was comparable to WT, and the loss of these different entities was not followed by any detectable increase in the free $F_1$ subcomplex level. To further investigate the importance of If1/Stf1 and free $F_1$ subcomplex interplay, we decided to investigate how loss of If1 impacts the phenotype of the $atp18\Delta$ strain lacking the $F_1F_0$-ATP synthase subunit i/j. This strain was previously characterized and presents a perturbed assembly and fragilized supramolecular organization of the $F_1F_0$-ATP synthase associated with a profound deficiency in enzyme activity. Loss of subunit i/j was also associated with an increased free $F_1$ subcomplex and oligomycin-insensitive activity (Vaillier et al, 1999; Wagner et al, 2010). The CN and BN-PAGE performed on the solubilized ATP synthase from total protein cell extracts of the $atp18\Delta$ mutant confirmed previous observations demonstrating that the levels of $F_1F_0$-ATP synthase oligomers were severely destabilized whereas the levels of free $F_1$ subcomplex were strongly increased (Fig. 3C). Interestingly, the isolated or combined loss of If1, inh1 in the $atp18\Delta$ strain we engineered, drastically reduced the level of the free $F_1$ subcomplex without affecting the levels of the monomeric $F_1F_0$-ATP synthase present in the $atp18\Delta$ strain (Fig. 3C–E). The predominant role of If1 on $F_1F_0$-ATP synthase stability compared to Stf1 was also observed on purified mitochondria where we confirmed that the isolated loss of If1 was sufficient to almost completely abolish the pH-dependent inhibition of $F_1F_0$-ATP synthase (Fig. EV1C). As expected, $F_1F_0$-ATP synthase dimers deficiency drastically impaired $atp18\Delta$ growth on medium containing glycerol (2%) a non-fermentable carbon source (Fig. 3F). Interestingly, the combined loss of the inhibitory peptides, which drastically reduced the level of the free

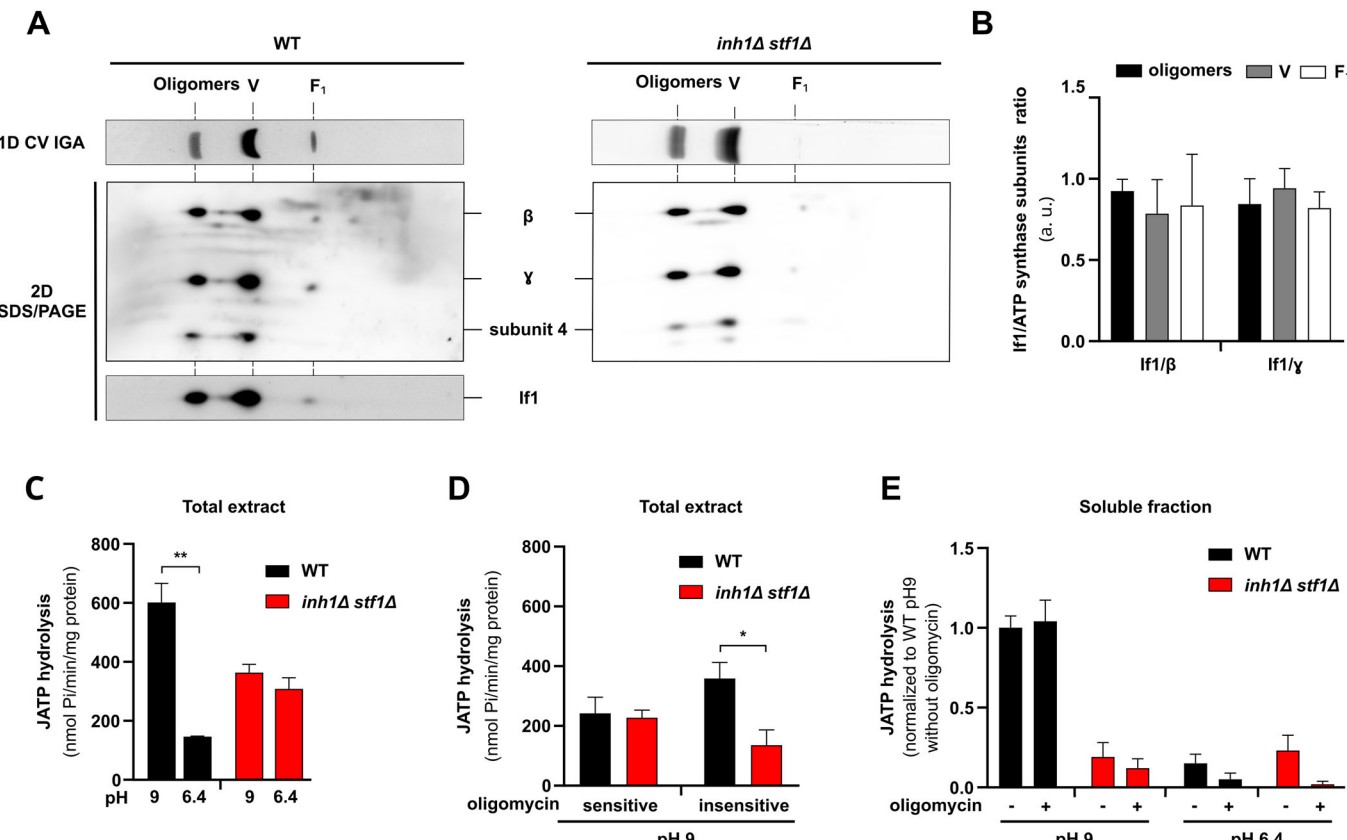

**Figure 2. If1 binds and inhibits ATP synthase oligomers, monomers and free F₁ subcomplexes.**

(A) Western blot following 2D-BN/SDS-PAGE performed with total cell extracts from WT and the *inh1Δ stf1Δ* grown on glycerol 2% rich medium. During the extraction and solubilization, the pH was conserved at 6.4 to preserve If1/Stf1 binding. (Representative of *n* = 3 independent experiments). (B) Densitometric quantification of western blot following 2D-BN/SDS-PAGE performed with total cell extracts from WT cells grown on glycerol 2% rich medium. For each F₁Fₒ-ATP synthase assembly, the western blot signal obtained with If1 signal was normalized to the β or γ subunit. (*n* = 3 independent experiments, 2-way ANOVA, error bars ± SEM). (C) Measurement of the ATP hydrolysis flux performed on total cell extracts from WT (black bars) and *inh1Δ stf1Δ* (red bars) grown on glycerol 2% rich medium by monitoring the ATP induced phosphate production over several minutes. Experiments were performed at pH 9.0 (inactive inhibitors) and pH 6.4 (active inhibitors). (*n* = 3 independent experiments, **p = 0.0022, unpaired t-test, error bars ± SEM). (D) Measurement of the ATP hydrolysis flux performed on total cell extracts from WT (black bars) and *inh1Δ stf1Δ* (red bars) grown on glycerol 2% rich medium by monitoring the ATP induced phosphate production over several minutes. Experiments were performed at pH 9.0 (inactive inhibitors) in absence or presence of oligomycin. (*n* = 3 independent experiments, *p = 0.0401, unpaired t-test, error bars ± SEM). (E) Measurement of the ATP hydrolysis flux performed on the soluble fraction purified from total cell extracts from WT (black bars) and *inh1Δ stf1Δ* (red bars) grown on glycerol 2% rich medium by monitoring the ATP induced phosphate production over several minutes. Experiments were performed at pH 9.0 (inactive inhibitors) and pH 6.4 (active inhibitors), in absence or presence of oligomycin. (*n* = 3 independent experiments, unpaired t-test, error bars ± SEM). Source data are available online for this figure.

F1 subcomplex, severely hampered the growth of *atp18Δ* mutant on glycerol (Fig. 3F,G).

## If1/Stf1 mitigate the impact of mitochondrial depolarizing stress on glycerol medium

To further evaluate the metabolic importance of If1/Stf1, we took advantage of the great metabolic flexibility of *S. cerevisiae* and characterized the phenotype of the *inh1Δ stf1Δ* strain grown under various carbon sources. To this end, we compared different carbon sources promoting respiro-fermentative (glucose 0.5% or galactose 2%) and non-fermentative conditions (glycerol 2% or lactate 2%). The respective capacity to metabolize these carbon sources and produce biomass were assessed using drop tests and growth curves (Fig. 4A,B). In parallel, the cellular respiration of yeast grown on different carbon sources was recorded using high-resolution respirometer O2K

oxygraph under endogenous conditions; or in presence of ethanol alleviating potential kinetic controls under endogenous, non-phosphorylating (triethyltin (TET)) and uncoupled (CCCP titration) states (Fig. 4C). These combined approaches assessing the growth of the *inh1Δ stf1Δ* strain in various metabolic conditions demonstrated that the loss of the inhibitory peptides did not affect growth (Fig. 4A,B). The cellular respiration assessed during exponential phase demonstrated that OXPHOS capacities are strongly adjusted in response to the carbon sources. As expected, the overall respiration rates as well as the part of the respiration devoted to ATP synthesis (respiration loss in response to triethyltin), were both increased from the highly glycolytic glucose medium to the highly oxidative lactate medium. Interestingly, the *inh1Δ stf1Δ* strain grown under galactose, glycerol or lactate, consistently exhibited a significantly higher respiration compared to the WT strain. These results indicate that the combined loss of If1/Stf1 is linked to an increased OXPHOS activity (Fig. 4C).

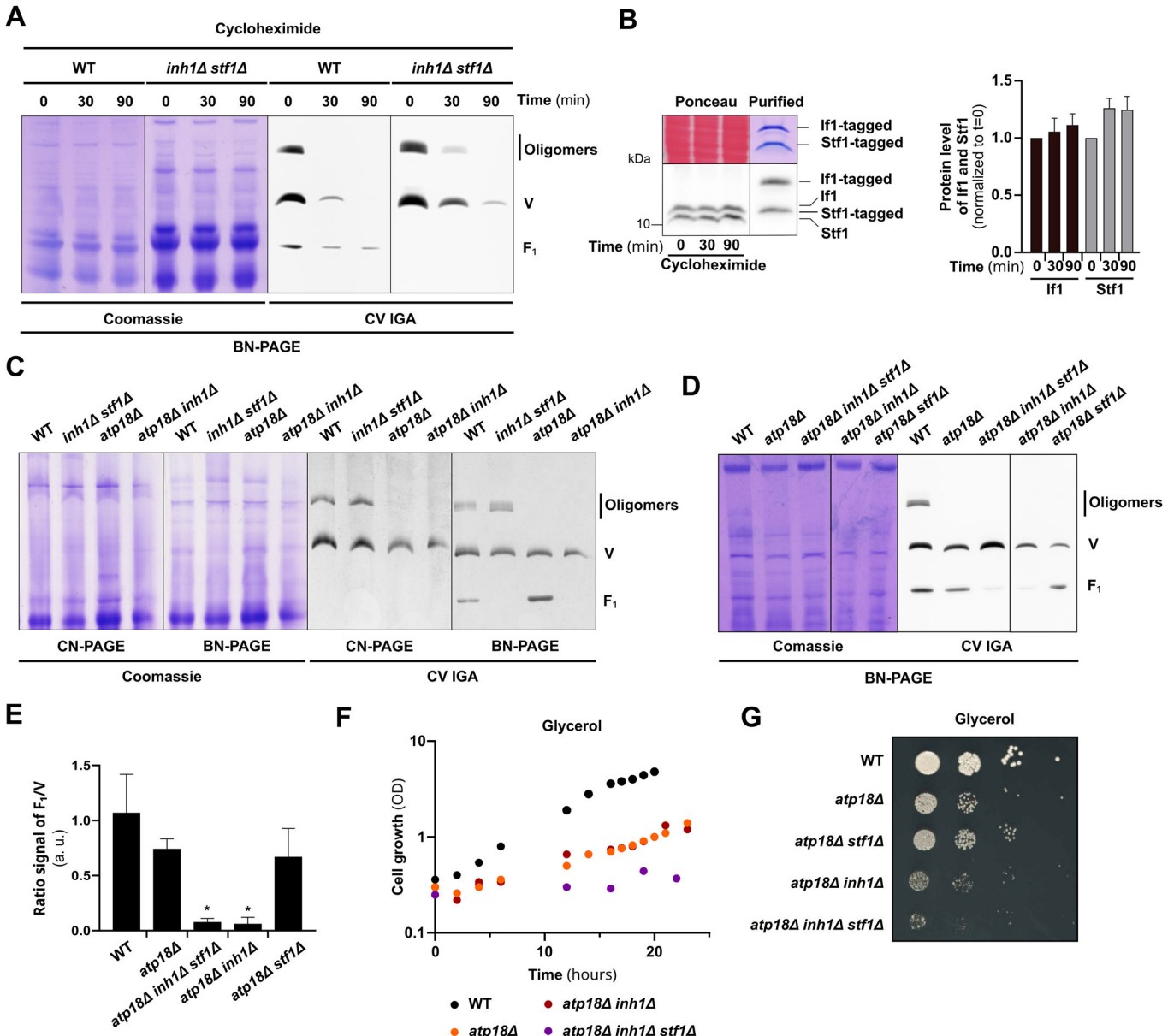

**Figure 3. If1 and Stf1 are specifically involved in free F₁ subcomplex stabilization.**

(A) BN-PAGE (3–12%) performed with total cell extracts, from WT and *inh1Δ stf1Δ* grown on glycerol 2% rich medium subjected to cycloheximide (250 µg/ml) during 0, 30, and 90 min. Samples were solubilized with digitonin at a digitonin-to-protein ratio of 1.5 g/g protein. The F₁F₀-ATP synthase assemblies were revealed by F₁F₀-ATP synthase hydrolytic in-gel activity (CV IGA). (Representative of $n = 3$ independent experiments). (B) Western blot (left) and densitometric analysis (right) of the relative abundance of If1 (black) and Stf1 (gray) from total cell extracts from WT grown on glycerol 2% rich medium subjected to cycloheximide (250 µg/ml) during 0, 30, and 90 min. Densitometric signals were normalized to the $t_0$-condition without cycloheximide. The ponceau staining as well as the Coomassie blue staining presented in the upper part demonstrate (i) that samples are evenly loaded and (ii) that the tagged If1 and Stf1 produced in vitro and used for relative quantification levels are equally loaded in the standard. The Coomassie blue staining of the purified peptides is also presented in Figs. 5A and EV1F. ($n = 3$ independent experiments, two-way ANOVA, error bars ± SEM). (C) CN and BN-PAGE (3–12%) performed with total cell extracts, from WT, *inh1Δ stf1Δ*, *atp18Δ* and *atp18Δ inh1Δ* cells grown on glycerol 2% rich medium solubilized with digitonin at a digitonin-to-protein ratio of 1.5 g/g protein. The F₁F₀-ATP synthase assemblies were revealed by F₁F₀-ATP synthase hydrolytic in-gel activity (CV IGA) (extended Coomassie and IGA from the data presented in Fig. 1D). (Representative of $n = 4$ independent experiments). (D) BN-PAGE (3–12%) performed with total cell extracts, from WT, *atp18Δ*, *atp18Δ inh1Δ stf1Δ*, *atp18Δ inh1Δ*, and *atp18Δ stf1Δ* grown on glycerol 2% rich medium solubilized with digitonin at a digitonin-to-protein ratio of 1.5 g/g protein. The F₁F₀-ATP synthase assemblies were revealed by F₁F₀-ATP synthase hydrolytic in-gel activity (CV IGA) (Representative of $n = 3$ independent experiments). (E) Densitometric quantification of the F₁ subcomplex (F₁) to monomer (V) signal ratio, from BN-PAGE performed with total cell extracts from WT, *atp18Δ*, *atp18Δ inh1Δ stf1Δ*, *atp18Δ inh1Δ*, and *atp18Δ stf1Δ* grown on glycerol 2% rich medium solubilized with digitonin at a digitonin-to-protein ratio of 1.5 g/g protein. The F₁F₀-ATP synthase assemblies were revealed by F₁F₀-ATP synthase hydrolytic in-gel activity (CV IGA). ($n = 3$ independent experiments, one-way ANOVA left to $*p = 0.0164$, $*p = 0.0339$, error bars ± SEM). (F) Growth of WT (black circles), *atp18Δ* (orange circles), *atp18Δ inh1Δ* (brown circles) and *atp18Δ inh1Δ stf1Δ* (purple circles) on glycerol 2% rich medium, following the optical density of the culture at 550 nm ($n = 3$ independent experiments). (G) Drop test performed on WT, *atp18Δ*, *atp18Δ stf1Δ*, *atp18Δ inh1Δ*, and *atp18Δ inh1Δ stf1Δ* grown on glycerol 2% rich medium (Representative of $n = 3$ independent experiments). Source data are available online for this figure.

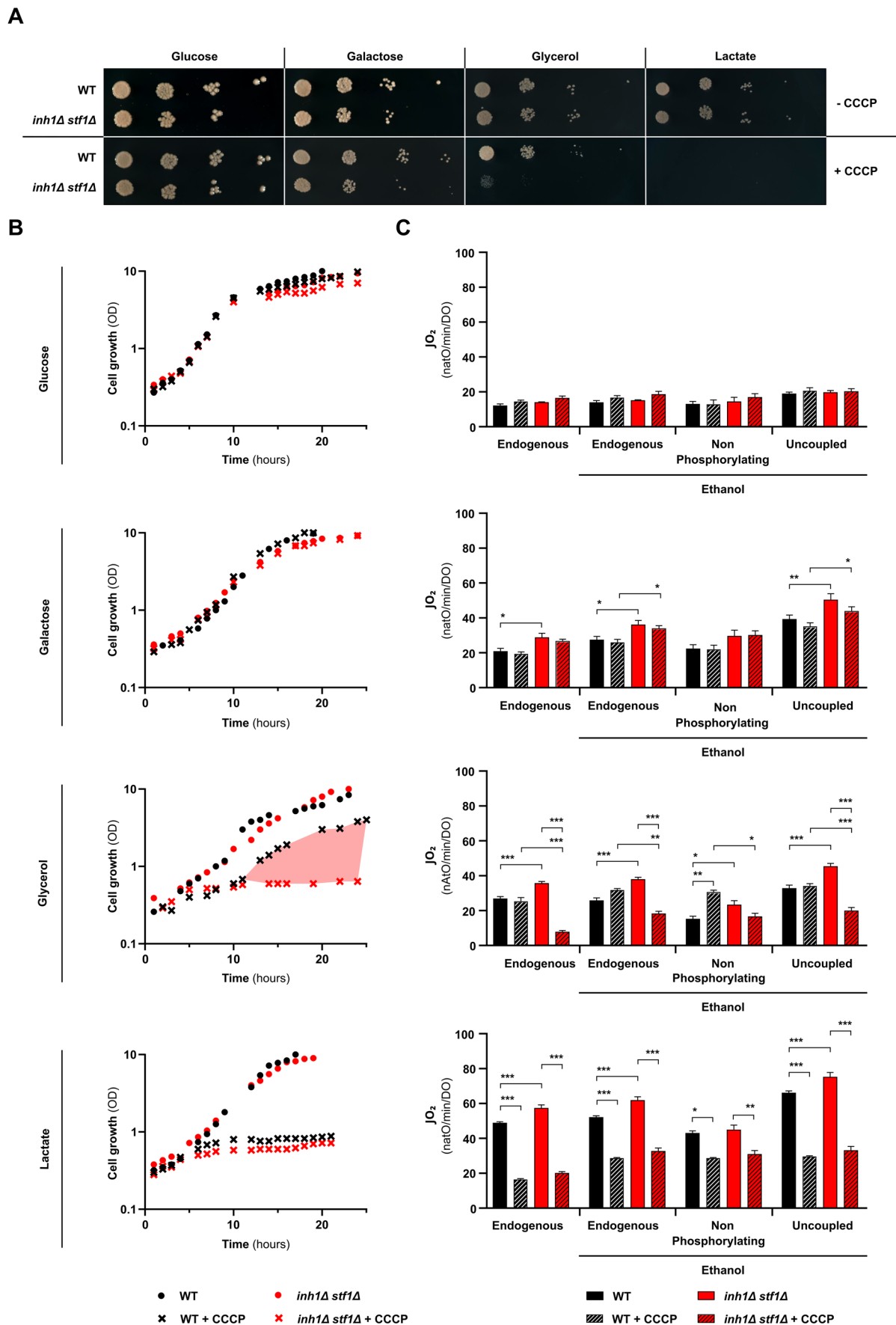

**Figure 4.  If1/Stf1 mitigate the impact of mitochondrial depolarizing stress on glycerol medium.**

(A) Drop test performed on WT and inh1Δ stf1Δ grown on different fermentable (glucose 0.5%, galactose 2%) and non-fermentable (glycerol 2%, lactate 2%) culture-rich media supplemented or not with CCCP (Representative of $n = 3$ independent experiments). (B) Growth of WT (black circles and crosses) and inh1Δ stf1Δ (red circles and crosses) cells on different fermentable (glucose 0.5%, galactose 2%) and non-fermentable (glycerol 2%, lactate 2%) culture-rich media supplemented (crosses) or not with CCCP (circles), following the optical density of the culture at 550 nm. ($n = 3$ independent experiments). (C) High-resolution respirometry performed on WT (black bars and hatched black bars in presence of CCCP) and inh1Δ stf1Δ cells (red bars and hatched red bars in presence of CCCP), collected during exponential phase on different fermentable (glucose 0.5%, galactose 2%) and non-fermentable (glycerol 2%, lactate 2%) culture-rich media, supplemented (hatched bars) or not (empty bars) with CCCP (9 h incubation). The oxygen consumption fluxes were normalized to optical density. Cellular respiration was measured under endogenous state and in the presence of ethanol, under endogenous, non-phosphorylating (triethyltin titration) and uncoupled (CCCP titration) states. ($n \geq 3$ independent experiments, from left to right and up and down *$p = 0.0477$, *$p = 0.0261$, *$p = 0.0497$, **$p = 0.0020$, *$p = 0.0238$, ***$p < 0.0001$, ***$p < 0.0001$, ***$p < 0.0001$, ***$p < 0.0001$, **$p = 0.0012$, ***$p < 0.0001$, **$p = 0.0010$, *$p = 0.0203$, *$p = 0.0301$, ***$p < 0.0001$, ***$p = 0.0006$, ***$p < 0.0001$, ***$p < 0.0001$, ***$p < 0.0001$, ***$p < 0.0001$, ***$p < 0.0001$, ***$p < 0.0001$, ***$p < 0.0001$, *$p = 0.0111$, **$p = 0.0080$, ***$p < 0.0001$, ***$p < 0.0001$, ***$p < 0.0001$, two-way ANOVA, error bars ± SEM). Source data are available online for this figure.

The enhanced OXPHOS activity of the inh1Δ stf1Δ strain prompted us to evaluate its capacity to respond to OXPHOS uncoupling stress induced by the mitochondrial depolarizing agent CCCP. To this end, yeast cultures were supplemented with the minimal CCCP concentration abolishing cellular respiration responses to triethyltin or CCCP validating that mitochondria are fully uncoupled (Fig. 4C). The CCCP treatment did not affect the growth of the inh1Δ stf1Δ or WT strains under fermentable carbon sources as glucose (0.5%) or galactose (2%) (Fig. 4A,B). However, the CCCP treatment differentially impacted the growth of the two strains under non-fermentable carbon sources i.e., glycerol (2%) or lactate (2%) (Fig. 4A,B). The mitochondrial uncoupling stress, under lactate (2%) medium, almost completely prevented the growth of both strains (Fig. 4A,B) causing severe deficiency of cellular respiration (Fig. 4C). In contrast, the inh1Δ stf1Δ and WT strains cultivated under glycerol (2%) strongly differ in their ability to tolerate the uncoupling stress (Fig. 4A,B). Our analyses clearly demonstrate that the If1/Stf1 inhibitors are essential to mitigate the impact of OXPHOS uncoupling stress on cell growth (Figs. 4A,B and EV1D) and respiration (Fig. 4C).

## Glycerol is a glyco-oxidative metabolic condition where the energy balance relies on ATP production from both glycolysis and OXPHOS

To further investigate the If1/Stf1-mediated mechanism involved in the OXPHOS uncoupling stress response, we first quantified If1 and Stf1 levels in different growth conditions. Western blot analyses performed on total protein extracts from WT yeast grown under various carbon sources demonstrated that the levels of both If1 and Stf1 normalized to the $F_1F_0$-ATP synthase subunit β, were almost doubled under the different non-fermentable conditions compared to glucose or galactose conditions (Fig. 5A,B). The change in If1/Stf1 levels were consistent with the increase in $F_1F_0$-ATP synthase levels observed in Native electrophoresis (Fig. EV1E). Our results also showed that the levels of both inhibitors were not impacted during the OXPHOS uncoupling stress (Fig. EV1F). The comparable expression of If1 and Stf1 under lactate and glycerol conditions indicated that the WT strain's resistance toward uncoupling stress in glycerol media was not merely due to changes in If1/Stf1 expression levels (Fig. 5A,B). We then decided to challenge the assumed metabolic homogeneity between glycerol or lactate conditions, commonly defined as non-fermentable carbon sources relying on OXPHOS activity. To determine the dependency of energy toward the OXPHOS or glycolytic produced ATP, we followed the growth of genetically engineered yeast mutants presenting altered glycolysis or OXPHOS

driven ATP production (Fig. 5C). The previously generated and characterized mutant yeast strains cdc19Δ and atp18Δ (Sprague, 1977; Vaillier et al, 1999), respectively suppressed of pyruvate kinase 1 and subunit i/j of ATP synthase protein. These mutants were selected to decipher the growth dependency toward these different energy producing pathways. The drop test analyses performed with these two strains (Figs. 5C and EV1G–I) and other mutants abolishing the ATP synthase assembly and activity (Fig. EV2A), nicely confirmed that the growth on glucose or galactose media almost exclusively relied on the glycolysis-driven ATP production (insensitive to OXPHOS ATP production defects). Conversely, the growth on lactate medium almost exclusively relied on the OXPHOS-driven ATP production (insensitive to defective glycolytic ATP production) (Figs. 5C and EV1G–I). However, the drop test experiment unexpectedly demonstrated that the growth on glycerol medium is hindered either by defects in OXPHOS or glycolysis-driven ATP production (Fig. 5C). This strongly confirms the ability of glycerol metabolism by-products such as glycerol-3-phosphate (G3P) and dihydroxyacetone phosphate (DHAP) to fuel both OXPHOS from Gut2p and glycolysis from the triose phosphate intermediates (Fig. 5D).

To further investigate the metabolic importance of the alternative glycolytic-driven ATP production in response to uncoupling stress, we measured, using HPIC method, the adenylic energy charge of inh1Δ stf1Δ and WT strains cultivated under glycerol medium (Fig. 5E). The adenylate energy charge measurement demonstrated that, in the WT strain, the glycolytic-driven ATP production pathway could preserve the cells' energy status from OXPHOS uncoupling stress. The fact that adverse energetic outcomes associated with the loss of If1/Stf1 only appeared under depolarization stress strongly suggests that the drop in adenylate energy charge is caused by the un-prevented ATP hydrolysis from the reversed mitochondrial $F_1F_0$-ATP synthase assemblies. Further analyses demonstrated that this acute energy crisis occurring within the first hour of treatment (Fig. 5E), affected cellular respiration at later time points (Fig. 4C). Interestingly, the respiration defects were associated with decreased levels and activities of complex III, IV, and V (Fig. 5F,G), but the OXPHOS deficiencies were not linked to mitochondrial genome deficiency (Fig. 5H) or loss of complex IV subunits (Fig. 5I). Altogether, our analyses demonstrate that the canonical role of If1/Stf1 in preventing ATP hydrolysis and safeguarding energy metabolism under mitochondrial depolarization stress is unmasked under the very specific glyco-oxidative metabolic condition, i.e., where yeast energy metabolism relies both on glycolysis and OXPHOS activities.

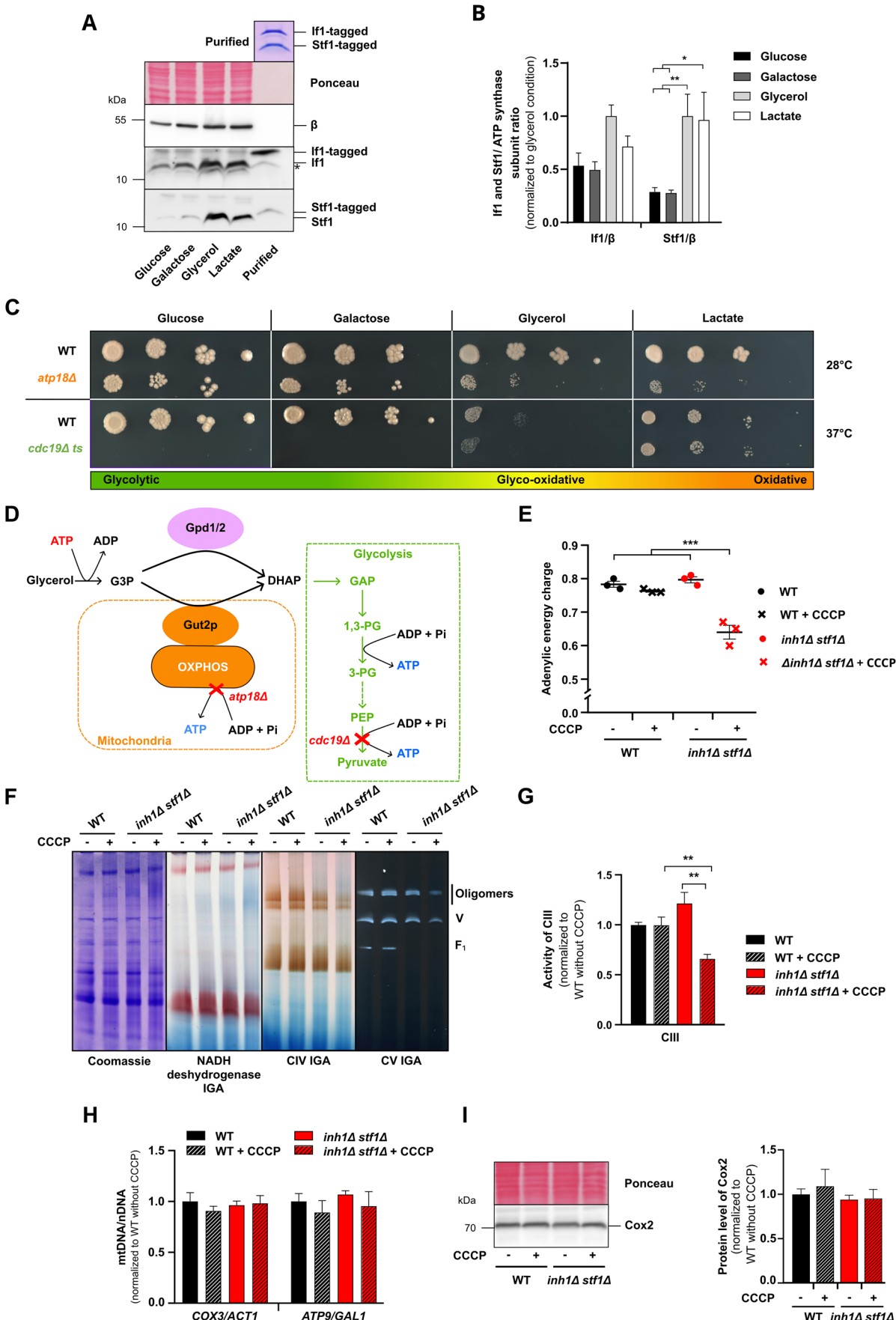

◄ **Figure 5. F₁F₀-ATP synthase peptide inhibitors activity is crucial to preserve energy metabolism under glyco-oxidative metabolism.**

(A) Western blot and (B) densitometric analysis of the relative abundance of If1 and Stf1 in regard to the F₁F₀-ATP synthase β subunit level. Denaturing electrophoresis was performed using total cell extracts from WT grown on different fermentable (glucose 0.5%, galactose 2%) and non-fermentable (glycerol 2%, lactate 2%) culture-rich media. Densitometric signals were normalized to the glycerol condition. The ponceau staining as well as the Coomassie blue staining presented in the upper part demonstrate (i) that samples are evenly loaded and (ii) that the tagged If1 and Stf1 produced in vitro and used for relative quantification levels are equally loaded in the standard. The Coomassie blue staining of the purified peptides is also presented in Figs. 3B and EV1F. ($n = 8$ independent experiments, from left to right $**p = 0.0074$, $**p = 0.0065$, $*p = 0.0122$, $*p = 0.0107$, two-way ANOVA, error bars ± SEM). (C) Drop test performed on WT, cdc19Δ thermosensitive (ts) mutant (37°) and atp18Δ mutant grown on different fermentable (glucose 0.5%, galactose 2%) and non-fermentable (glycerol 2%, lactate 2%) culture minimum media. (Representative of $n = 3$ independent experiments). (D) Scheme of the main metabolic pathways involved in ATP/ADP maintenance characterizing the so-called glyco-oxidative metabolism observed under glycerol 2% conditions. (E) HPIC quantification of adenylate energy charge in WT (black) and inh1Δ stf1Δ (red) cells grown in the absence (circles) or presence (crosses) of CCCP (1 h incubation) on glycerol 2% rich medium. ($n = 3$ independent experiments, from left to right $***p = 0.0003$, $***p = 0.001$, $***p = 0.0002$, one-way ANOVA, Error bars ± SEM). (F) BN-PAGE (3–12%) performed with total cell extracts from WT and inh1Δ stf1Δ grown on glycerol 2% rich medium solubilized with digitonin at a digitonin-to-protein ratio of 1.5 g/g protein. NADH dehydrogenase, complex IV (CIV) and F₁F₀-ATP synthase (CV) in-gel activities (IGA) were performed. (Representative of $n = 3$ independent experiments). (G) Enzymatic activity of the respiratory chain complex III (CIII) performed on total cell extracts from WT and inh1Δ stf1Δ cultivated on glycerol 2% rich medium, in the presence or absence of CCCP (6 h incubation). All the fluxes were normalized to WT condition without CCCP ($n ≥ 9$ independent experiments, from left to right $**p = 0.0027$, $**p = 0.0022$, unpaired t-test, error bars ± SEM). (H) Quantitative PCR quantification of the mitochondrial DNA to nuclear DNA ratio performed on total DNA extracted from WT and inh1Δ stf1Δ grown on glycerol 2%, in the presence (hatched bars) or absence (solid bars) of CCCP (6 h incubation) ($n = 3$ independent experiments, 2-way ANOVA, error bars ± SEM). (I) Western blot (left) and densitometric (right) analysis of the relative abundance of Cox2, a complex IV subunit. Denaturing electrophoresis was performed using total cell extracts from WT and inh1Δ stf1Δ grown on glycerol 2% rich medium in the presence or absence of CCCP (6 h incubation). Immunodetected signals were normalized to ponceau signal ($n = 3$ independent experiments, one-way ANOVA, error bars ± SEM). Source data are available online for this figure.

## The If1/Stf1 inhibitors as well as F₁F₀-ATP synthase-free F₁ subcomplex are both dispensable for the viability of ρ⁻/° yeasts

According to several reports, the ATPase activity of the free F₁ subcomplex coupled to the electrogenic activity of the adenine nucleotide translocator (ANT), is crucial to support mitochondrial membrane potential and growth of yeast lacking their mitochondrial genome (ρ⁻/° cells) (Clark-Walker, 2007; Giraud and Velours, 1997; Kominsky et al, 2002). The improved growth capacity of ρ⁻/° cells lacking If1 also strongly endorsed the importance of the free F₁ subcomplex ATPase activity in supporting the growth of mtDNA-deprived cells (Liu et al, 2021). These previous works supporting that free F₁ subcomplexes could be more active in the absence of If1 in the context of ρ⁻/°, would contrast with the interdependence between If1/Stf1 and free F₁ subcomplex we observed (Figs. 1 and 2). Therefore, we decided to characterize the interplay between If1/Stf1 and free F₁ subcomplex in the context of ρ⁻/° cells. First, during the stationary growth phase of WT and inh1Δ stf1Δ, we analyzed the proportion of cells undergoing ρ⁻/° conversion. In line with previous works, we observed that the loss of If1/Stf1 activity was in fact favoring the loss of mitochondrial genome and the conversion of yeast into ρ⁻/°cells (Fig. 6A). Then, we generated stable WT and inh1Δ stf1Δ ρ⁻/° cells and observed that the If1/β ratio was unchanged in WT ρ⁻/° whereas the Stf1/β ratio, while not significant, tended to be reduced (Fig. 6B). While favoring the conversion into ρ⁻/° cells (Fig. 6A), the loss of If1/Stf1 did not impact the ρ⁻/° cells' growth rate under fermentative conditions (Fig. 6C). Moreover, our data demonstrated that the overall ATP hydrolysis activity (Fig. 6D) as well as the free F₁ subcomplex expression (Fig. 6E) were barely detected in total extracts from WT and inh1Δ stf1Δ ρ⁻/° cells. Interestingly, BN-PAGE experiments performed on highly concentrated extracts from ρ⁻/° cells confirmed that the free F₁ subcomplex levels were, like in the ρ⁺ context (Fig. 1D), severely reduced in the inh1Δ stf1Δ ρ⁻/° compared to the WT ρ⁻/° cells (Fig. 6F). Collectively, our findings suggest that the deletion of If1/Stf1 is beneficial for ρ⁻/° cells, but they also refute hypotheses supporting that this effect could be tightly linked to an activation of the F₁ subcomplex driven ATP hydrolysis. Instead, our results demonstrated that the loss of If1/Stf1 together with the free F₁ subcomplex favored the conversion into ρ⁻/° and did not affect the growth of ρ⁻/° cells on fermentable carbon sources (Fig. 6).

# Discussion

The present work confirmed that, in contrast to their mammalian homolog If1, the yeast If1/Stf1 inhibitors are dispensable for the biogenesis and stability of F₁F₀-ATP synthase dimers and provides first evidence that If1/Stf1 are not required for the stability of F₁F₀-ATP synthase oligomers. The resolution of the F₁F₀-ATP synthase tetramers' structures by Cryo-EM, demonstrating that If1 dimers could bridge and likely stabilize adjacent dimers (Gu et al, 2019; Pinke et al, 2020; Mühleip et al, 2021), has supported the previously proposed role of If1 in the supramolecular organization of the mammalian F₁F₀-ATP synthase (Cabezón et al, 2000). The role of mammalian If1 in F₁F₀-ATP synthase supramolecular organization was recently strengthened, by single-molecule tracking microscopy and native electrophoresis performed on If1 knock-in and knock-out cell lines (Romero-Carramiñana et al, 2023; Weissert et al, 2021). In contrast, the lack of cryo-EM structures of yeast F₁F₀-ATP synthase oligomers larger than dimers (Hahn et al, 2016), along with previously characterized differences in If1 dimerization and function, has cast doubts on the potential implication of If1/Stf1 inhibitors in the F₁F₀-ATP synthase supramolecular organization in yeast (Cabezon et al, 2002; Hong and Pedersen, 2002; Le Breton et al, 2016). Our work confirms previous findings showing that the levels of F₁F₀-ATP synthase dimers remained unchanged in inh1Δ stf1Δ (Dienhart et al, 2002), but also demonstrates that loss of these inhibitory peptides does not preclude the assembly and levels of higher molecular weight F₁F₀-ATP synthase oligomers (Fig. 1C). Instead, structural and functional investigations consistently demonstrated that If1 is required to maintain the levels of the free F₁ subcomplex, which is functionally characterized by its oligomycin insensitivity (Figs. 1 and 2) (Wittig et al, 2007). Our findings also support previous works demonstrating that ATP

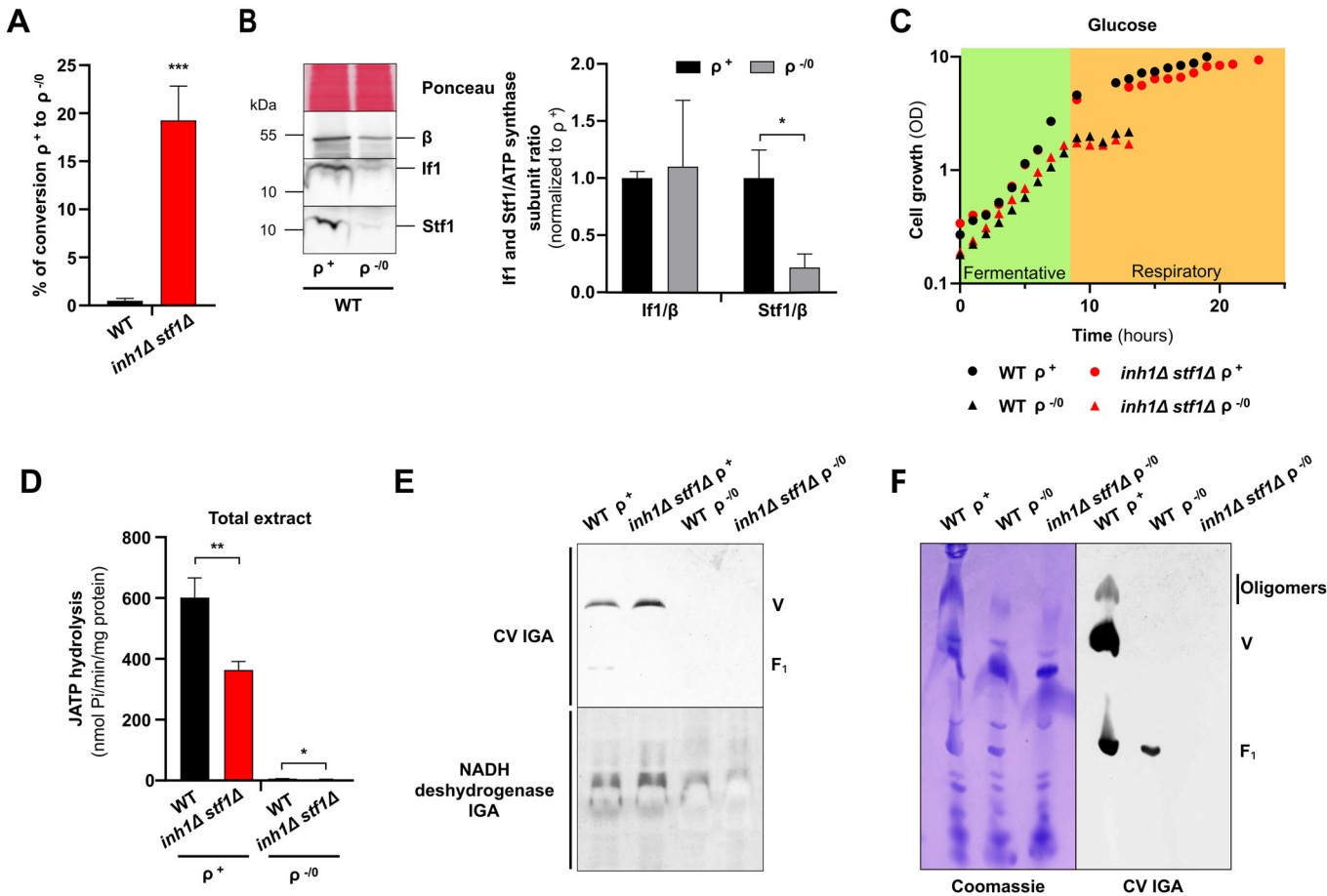

**Figure 6.  The If1/Stf1 inhibitors as well as F₁F₀-ATP synthase-free F₁ subcomplex are both dispensable for the viability of ρ⁻/° yeast.**

(A) Percentage of WT and *inh1Δ stf1Δ* cells that spontaneously lost their mitochondrial genome ρ⁻/°. ($n = 8$ independent experiments, ***$p = 0.0001$ unpaired t-test, error bars ± SEM). (B) Western blot (left) and densitometric analysis (right) of the relative abundance of If1 and Stf1 in regards to the F₁F₀-ATP synthase β subunit level. Denaturing electrophoresis was performed on total cell extracts from WT (black) and WT ρ⁻/° (gray) grown on glucose 0.5% rich medium. ($n = 3$ independent experiments, *$p = 0.0454$, unpaired t-test, error bars ± SEM). (C) Growth of WT (black circles) or WT ρ⁻/° (black triangles), *inh1Δ stf1Δ* (red circles), and *inh1Δ stf1Δ* ρ⁻/° (red triangles) cell on glucose 0.5% rich medium, following the optical density of the culture at 550 nm. ($n = 3$ independent experiments). (D) Measurement of the ATP hydrolysis flux performed on total cell extracts from WT (black bars) and *inh1Δ stf1Δ* (red bars) and their ρ⁻/° variants, grown on glucose 0.5% rich medium by monitoring the ATP induced phosphate production over several minutes. Experiments were performed at pH 9.0 (inactive inhibitors). ($n = 3$ independent experiments, from left to right **$p = 0.0022$, *$p = 0.0245$, unpaired t-test, error bars ± SEM). (E) BN-PAGE (3–12%) performed with total cell extracts from WT and *inh1Δ stf1Δ* and their respective ρ⁻/°- variants, grown on glucose 0.5% rich medium. Cell extracts were solubilized with digitonin at a digitonin-to-protein ratio of 1.5 g/g protein. NADH dehydrogenase and F₁F₀-ATP synthase (CV) in-gel activities (IGA) were performed. (Representative of $n = 3$ independent experiments). (F) BN-PAGE (3–12%) performed with highly concentrated total cell extracts from WT, WT ρ⁻/° and *inh1Δ stf1Δ* ρ⁻/° grown on glucose 0.5% rich medium. Cell extracts were solubilized with digitonin at a digitonin-to-protein ratio of 1.5 g/g protein. The F₁F₀-ATP synthase assemblies were revealed by ATP synthase hydrolytic in-gel activity (CV IGA) (Representative of $n = 3$ independent experiments). Source data are available online for this figure.

synthase dimers can form rows and induce membrane curvature just from the intrinsic shape of the dimer without the need for If1 connecting neighboring dimers together (Davies et al, 2012; Anselmi et al, 2018; Blum et al, 2019).

The free F₁ subcomplex observed under native PAGE or density gradients used to be considered as an artifact resulting from the destabilization of fully assembled F₁F₀-ATP synthase (monomers and oligomers) by the action of detergent or improper sample preparation and storage (Ackerman and Tzagoloff, 1990; Jänsch et al, 1996; Wittig et al, 2007). The native PAGE and solubilization techniques development combined with the use of milder detergents such as digitonin have since convincingly demonstrated that the free F₁ subcomplexes were stable assembly intermediates

formed independently from the ATP synthase F₀ sectors (Li et al, 2012; Nijtmans et al, 1995). The native PAGE and oligomycin-sensitive enzymatic analyses, have independently demonstrated that the ratio between the free F₁ subcomplex and the fully assembled F₁F₀-ATP synthase was impacted neither by the use of mild detergent nor by the BN-PAGE approach. Our findings support that the free F₁ subcomplex is not a degradation by-product resulting from digitonin treatment since no F₁ subcomplex could be detected in the digitonin-solubilized membrane fraction containing fully assembled F₁F₀-ATP synthases (Fig. 1C–E). Instead, our work strongly supports previous reports claiming that the free F₁ subcomplex is a stable intermediate assembly. This hypothesis is also strengthened by the enhanced levels of free F₁ we

observed in yeast presenting aberrant ATP synthase assembly such as atp18Δ (Fig. 3C) and ρ⁻/° (Fig. 6E,F), corroborating independent observations obtained in different model organisms (Carrozzo et al, 2006; Wittig et al, 2006, 2007; Mourier et al, 2014; He et al, 2018). Our quantitative analyses substantiate independent reports demonstrating that If1 binding affinity and inhibitory capacities are greater than Stf1 (Cabezon et al, 2002; Venard et al, 2003). Interestingly, we demonstrated that If1 evenly binds the various ATP synthase assemblies (Fig. 2A,B) and can very efficiently inhibit their ATP hydrolysis activity in a characteristic pH-dependent manner (Fig. 2C–E).

Several independent studies have previously demonstrated that the maintenance of mitochondrial membrane potential and the ADP/ATP translocator were essential for the survival and growth of yeast presenting long range deletion (ρ⁻) or complete loss of their mitochondrial genome (ρ°) (Dupont et al, 1985; Kováčová et al, 1968; Subík et al, 1972). This specific sensitivity prompted scientists to hypothesize that the $F_1$ subcomplex driven ATP hydrolysis together with the electrogenic activity of the ADP/ATP translocator could be key in maintaining the proton electrochemical potential across the mitochondrial inner membrane and therefore essential for the growth and survival of ρ⁻/° yeast (Giraud and Velours, 1997; Chen and Clark-Walker, 1999; Clark-Walker, 2007). Our analyses demonstrated that the loss of If1/Stf1 associated with drastic loss of $F_1$ subcomplex does not prevent the yeast to undergo ρ⁻/° conditions nor impact their growth under fermentative carbon source (Fig. 6). Consequently, our results corroborate original analyses from Tzagaloff and Schatz (Tzagoloff et al, 1975) demonstrating that the loss of subunits forming the $F_1$ sector did not preclude the conversion into ρ⁻/° cells.

For endogenous peptide inhibitors, the free $F_1$ subcomplex is a priority target because of its potential toxicity for the cell's energy metabolism. In contrast to the fully assembled $F_1F_0$-ATP synthase, the toxicity of the free $F_1$ subcomplex stems from its lack of thermodynamic feedback inhibition by the proton electrochemical potential. The novel fIf1/Stf1-mediated mechanism of action identified in the present work, elegantly circumvents any potentially uncontrolled ATP hydrolysis from the yeast-free $F_1$ subcomplex. Our present work demonstrates that in the presence of If1/Stf1 the free $F_1$ subcomplex is tightly regulated and inhibited, whereas in the absence of If1 the free $F_1$ subcomplex is not maintained (Figs. 1 and 2). Altogether, our data suggest that the If1 binding to the free $F_1$ subcomplex is not only preventing potential toxic ATP hydrolysis, but also stabilizing this assembly intermediate (Fig. 3A). This peculiar interplay between If1 and the yeast-free $F_1$ subcomplex is in agreement with the parallel increase in If1 and ATP synthase subassemblies we previously characterized in cardiac-specific knockouts developing a progressive cardiomyopathy associated with mitochondrial genome expression deficiency (Mourier et al, 2014).

Interestingly, the loss of If1/Stf1 inhibition on the fully assembled $F_1F_0$-ATP synthase was associated with a mild stress increasing cellular respiration capacity, but without affecting the growth rate (Fig. 4). However, the CCCP-induced OXPHOS uncoupling stress unmasks the crucial role played by If1/Stf1 under a very specific metabolic condition associated with the glycerol carbon source. The systematic comparison of cellular respiration of cells growing under fermentable (glucose 0.5%, galactose 2%) and non-fermentable (glycerol 2%, lactate 2%) carbon sources demonstrated that mitochondrial respiration alone was insufficient to comprehend the unique metabolic specificity of cells metabolizing glycerol (Fig. 4). In contrast to the glucose or lactate conditions presenting striking differences in their respirations, it was impossible to discriminate, on the basis of their endogenous respiration, yeast grown on galactose (non-fermentable) or glycerol (fermentable) (Fig. 4). This observation prompted us to develop a new screening tool using genetically modified yeast presenting defective mitochondrial or glycolytic-driven ATP production to decipher the respective implication of the two pathways in energy balance and yeast growth (Fig. 5C). This strategy demonstrated that in contrast to lactate medium where yeast growth was exclusively dependent on the mitochondrial pathway, growth on glycerol also relied on the pyruvate kinase 1 (Cdc19) glycolysis activity. This observation demonstrated that glycerol, in contrast to lactate, is not a strict 'non-fermentable' carbon source strictly relying on mitochondrial energy metabolism. This observation corroborates recent hypotheses based on transcriptomic and metabolomic analyses and challenging the classification of glycerol as a 'non-fermentable' carbon source (Aßkamp et al, 2019; Galkina et al, 2022; Xiberras et al, 2019). Assessing the relative dependence of cell energy metabolism on glycolysis and oxidative phosphorylation was more pertinent for understanding the role of If1/Stf1 in OXPHOS uncoupling stress in glycerol, than simply characterizing metabolism through respiro-fermentative properties. Accordingly, the specific If1/Stf1 dependency of cells under glycerol, redefined here as a glyco-oxidative metabolic condition, suggests that preventing mitochondrial ATP hydrolysis is crucial for cell growth only when energy metabolism evenly relies on both glycolysis and OXPHOS processes (Fig. 5C–E). In contrast, the unrepressed ATPase activity did not affect cell growth on highly glycolytic conditions (glucose or galactose) and conversely could not be compensated under highly oxidative metabolic conditions (lactate). We believe that this new approach deciphering the respective roles of glycolysis and OXPHOS in cell energy balance applied to the mammalian context could help understand the intriguing role of If1 in numerous tumors undergoing hypoxic stress and overexpressing this inhibitory peptide (Sánchez-Aragó et al, 2013; Sgarbi et al, 2018).

# Methods

**Reagents and tools table**

| Reagent/Resource | Reference or Source | Identifier or Catalog Number |
| --- | --- | --- |
| **Experimental models** | | |
| D273-711 10B/A/H/U | Paul et al, 1989 | |
| BY4741 | Euroscarf | Y00000 |
| BY4743 | Euroscarf | Y20000 |
| BY4741 atp18Δ | Euroscarf | Y06068 |
| BY4741 cdc19Δ | Euroscarf | Y40615 |
| BY4742 atp5Δ | Euroscarf | Y13657 |
| BY4742 atp7Δ | Euroscarf | Y14865 |
| BY4742 atp14Δ | Euroscarf | Y15205 |

| Reagent/Resource | Reference or Source | Identifier or Catalog Number |
|---|---|---|
| **Recombinant DNA** | | |
| pFA6a-His3MX6 | Addgene | 41596 |
| pUG-natNT2 | Addgene | 110922 |
| **Antibodies** | | |
| Mouse Pgk1 22C5D8 (monoclonal) | CiteAb | 459250 |
| Mouse MTCO2 (Cox2) (monoclonal) | Thermo Fisher | 12C4F12 |
| Rabbit subunit β (polyclonal) | This study | |
| Rabbit subunit γ (polyclonal) | This study | |
| Rabbit subunit 4(polyclonal) | This study | |
| Rabbit subunit I (polyclonal) | This study | |
| Rabbit If1 (polyclonal) | This study | |
| Rabbit Stf1 (polyclonal) | This study | |
| Peroxidase Goat anti-rabbit IgG (polyclonal) | Jackson ImmunoResearch | AB_2313567 |
| Peroxidase Goat anti-mouse IgG (polyclonal) | Jackson ImmunoResearch | AB_2338504 |
| **Oligonucleotides and other sequence-based reagents** | | |
| PCR primers | This study | Methods |
| **Chemicals, Enzymes and other reagents** | | |
| NativePAGE™ Bis-Tris Gels, 3 to 12% | ThermoFisher | BN1001BOX |
| NuPAGE™ 4 à 12%, Bis-Tris gels | ThermoFisher | NP0321BOX |
| Amersham™ Protran® Western blotting membranes, nitrocellulose | Dutscher | 10600004 |
| PageRuler™ Plus | ThermoFisher | 26619 |
| Zymolyase®-20T | nacalai tesque® | 07663-04 |
| Complete EDTA-free™ | Roche | 11836170001 |
| centrifugal Concentrator Corning® Spin-X® | MERCK | CLS431478 |
| qPCRBIO SyGreen Blue Mix Lo-ROX | Eurobio® | PB20.15-05 |
| Clarity Western ECL Substrate | Bio-Rad | 1705061 |
| **Software** | | |
| ImageJ | https://imagej.nih.gov/ij/index.html | |
| GraphPad Prism 8.0 | https://www.graphpad.com/ | |
| **Other** | | |
| Oroboros Instruments | https://www.oroboros.at/ | |
| Amersham ImageQuant™ | | |
| Spectrophotometer Jasco V-760 | https://www.jascofrance.fr/ | |
| DC™ protein assay kit | Bio-Rad | 5000111 |

## Yeast strains

The *Saccharomyces cerevisiae* strain used in this study is the strain D273-10B/A/H/U (*MAT α, met6, ura3, his3*) referred to as the wild type (WT) (Paul et al, 1989) and strain BY4741 Euroscarf (*MAT α; his3Δ1; leu2Δ0; met15Δ0; ura3Δ0*). The mutants D273 were obtained by homologous recombination of the following deletion cassette in the wild type strain: D273 *inh1Δ* (*MAT α, met6, ura3, his3, inh1::HIS3-kanMX6*), D273 *stf1Δ* (*MAT α, met6, ura3, his3, stf1::Nat^R*), D273 *inh1Δ stf1Δ* (*MAT α, met6, ura3, his3, stf1::Nat^R, inh1::HIS3-KanMX6*), D273 *atp18Δ* (*MAT α, met6, ura3, his3, atp18::HIS3-KanMX6*), D273 *atp18Δ inh1Δ* (*MAT α, met6, ura3, his3, atp18::Nat^R, inh1::HIS3-kanMX6*), D273 *atp18Δ stf1Δ* (*MAT α, met6, ura3, his3, atp18::HIS3-kanMX6, stf1::Nat^R*), D273 *atp18Δ inh1Δ stf1Δ* (*MAT α, met6, ura3, his3, atp18::NatR, inh1::HIS3-kanMX6, stf1::Kan^R*). The plasmids used for the cassette *HIS3-kan*MX6 and nourseothricin resistance (*Nat^R*) were pFA6a-*His3*MX6 and pUG-natNT2 respectively. The oligonucleotides used are listed in Table EV1. The generated mutants were validated by PCR and Western blot. D273 ρ$^{-/o}$ strain was obtained after growing cells on rich medium containing glucose 2% for two days before spreading on rich medium with 0.5% of glucose. The ρ$^{-/o}$ colonies were identified and counted by comparing to a replica plate containing glycerol rich medium where only the ρ$^+$ colonies were able to grow. The wild type and mutants strains BY4741 (*MAT a; his3Δ1; leu2Δ0; met15Δ0; ura3Δ0*), BY4742 (*MATα, lys2Δ0, ura3Δ0, his3Δ1, leu2Δ0*) and BY4743 (*MATa/MATα; his3Δ1/his3Δ1; leu2Δ0/leu2Δ0; LYS2/lys2Δ0; met15Δ0/MET15; ura3Δ0/ura3Δ0*) were provided by Euroscarf: BY4741 *atp18Δ* (*MAT a; his3Δ1; leu2Δ0; met15Δ0; ura3Δ0; YML081c-a::kanMX4*), BY4741 *cdc19Δ* (*MAT a; his3Δ1; leu2Δ0; met15Δ0; ura3Δ0; cdc19-1::kanMX4* (lethal at 37 °C on glucose medium)), BY4742 *atp5Δ* (*MATα, lys2Δ0, ura3Δ0, his3Δ1, leu2Δ0, atp5::KAN^R*), BY4742 *atp7Δ* (*MATα, lys2Δ0, ura3Δ0, his3Δ1, leu2Δ0, atp7::KAN^R*), BY4742 *atp14Δ* (*MATα, lys2Δ0, ura3Δ0, his3Δ1, leu2Δ0, atp14::KAN^R*).

## Yeast growth and media

Cells were grown aerobically at 28 °C with shaking at 180 rpm. Growth was followed by measuring the optical density at 550 nm using a Jasco V-760 spectrophotometer. The composition of rich medium was: 0.1% (w/v) $KH_2PO_4$ pH 5.5, 1% (w/v) yeast extract, 0.12% (w/v) $(NH_4)_2SO_4$, 2% or 0.5% (w/v) carbon source and for solid medium 2% (w/v) bacto agar was added. For some drop test, cells were grown in the following minimum medium: 0.1% (w/v) $KH_2PO_4$ pH 5.5, 0.175% (w/v) yeast nitrogen base w/o amino acids and ammonium sulfate, 0.5% (w/v) $(NH_4)_2SO_4$, 0.2% (w/v), 2% or 0.5% (w/v) carbon source, 2% (w/v) bacto agar minimum medium, 0.2% (w/v) casein hydrolysate, 100 mg/l leucine, 20 mg/l histidine, 20 mg/l methionine, 20 mg/l uracil. A filtered solution of casein and amino acids was added to the sterilized medium. Different carbon sources were used: D,L-lactic acid, glycerol, D(+)-galactose, or D(+)-glucose. The carbon source and the type of medium (rich or minimum) selected for each experiment is indicated in the legends.

The uncoupler agent CCCP was added in the liquid culture medium, at 28 °C, a few minutes before inoculating the cells at 0.1 $OD_{550nm}$ for growth curves and at 1 $OD_{550nm}$ for HPIC experiments. For solid culture medium CCCP was added in the tepid medium just before solidification of agar and used within one day. A CCCP concentration titration (between 1.25 and 7.5 μM) was performed for each experiment and condition. The inhibitor of cytosolic translation, cycloheximide (250 μg/ml) was added in the culture medium, at 2 $OD_{550nm}$ culture. Cells were harvested during exponential growth.

## Mitochondria preparation

Yeast cells grown in the presence of 2% (w/v) lactate were collected during the exponential growth phase and mitochondria were prepared by enzymatic digestion of the cell wall with Zymolyase®-20T (nacalai tesque®, reference 07663-04) according to (Guérin et al, 1979).

## High-resolution oxygen consumption measurement on yeast cells and isolated mitochondria

Oxygen consumption of cells harvested during the exponential growth phase on different culture media was measured at 28 °C using an Oxygraph-2k (OROBOROS INSTRUMENTS, Innsbruck, Austria). Oxygen consumption rates of cells were measured under endogenous state and in presence of 85 mM ethanol substrate under endogenous conditions, under the non-phosphorylating state with addition of 25 μM TET and under the uncoupled state by successive addition of CCCP titration (around 2.5 μM). When necessary, cells were diluted with the culture medium.

## Protein extraction

100 units of $OD_{550nm}$ were harvested, pelleted and washed with cold water before being broken by vigorous shaking for 4 min in 250 μl of extraction buffer containing 10 mM Bis-Tris-HCl pH 6.4, 1 mM EDTA and a mixture of protease inhibitors (Complete EDTA-free™, Roche) with an equal volume of glass beads (0.4 mm diameter). Protein concentrations were then determined using the DC assay according to manufacturer's instructions (Bio-Rad). To detect the free $F_1$ subcomplex in $\rho^{-/o}$, we obtained concentrated solubilized cell protein extracts using a centrifugal Concentrator (Corning® Spin-X® UF 500 μL, molecular weight cut-off of 100 kDa). The supernatant was collected after centrifugation at $12,000 \times g$ for 10 min at 4 °C. To obtain the membrane and soluble fractions, the total cell extract was centrifuged at $30,000 \times g$ during 30 min at 4 °C. The supernatant (containing the soluble fraction) and pellet (containing the membrane fraction), were collected and subjected to protein quantification. The pellet was resuspended in a volume of extraction buffer equivalent to that of the harvested supernatant.

## Enzymatic activities determination

Cytochrome c reductase (complex III) activity was determined by the absorbance at 550 nm of reduced cytochrome c in the following buffer: 50 mM $KH_2PO_4$ pH 7.4, 0.5 mM KCN, 10 mM succinate, 10 mM G3P and 200 μg/ml bovine heart cytochrome c. The

specificity of the assay was validated by the addition of 0.5 μM antimycin A. Cytochrome c reductase (complex III) activity being defined as the antimycin A sensitive flux. Enzymatic activity measurements were performed with a Jasco V-760 spectrophotometer on 250 μg protein in a quartz spectrophotometer cell at 28 °C with stirring at 1000 rpm.

The hydrolytic activity of ATP synthase (complex V) was determined with 0.8 mg protein cell extracts at 28 °C with shaking at 750 rpm with a thermomixer in the following buffer: 75 mM triethanolamine pH 9.0 or pH 6.4, 5 mM $MgCl_2$ with 2 μg/ml alamethicin. The reaction was initiated with the addition of 5 mM ATP and an aliquot was collected every 2 min (or 6 min for $\rho^{-}/^o$) and added to the following solution: 0.38 M sulfuric acid, 5 μM ammonium heptamolybdate tetrahydrate, 29 μM iron(II) sulfate heptahydrate. The Pi product was quantified following changes in the absorbance assessed at 750 nm. The same experiment was performed after a 2-min pre-incubation with 19 μg/ml oligomycin to assess the oligomycin-insensitive ATP hydrolysis flux. The hydrolytic activity of ATP synthase (complex V) on purified mitochondria (0.035 mg of mitochondrial protein) was determined at 28 °C with shaking at 750 rpm in an Oroboros chamber in the following buffer: 75 mM triethanolamine pH 9.0 or 6.4, 5 mM $MgCl_2$ with 1 μg/ml alamethicin. The reaction was initiated with the addition of 5 mM ATP. Every 30 s an aliquot was collected and added to the following solution: 0.38 M sulfuric acid, 5 μM ammonium heptamolybdate tetrahydrate, 29 μM iron(II) sulfate heptahydrate. The Pi product was quantified following changes in the absorbance assessed at 750 nm. The same experiment was performed after a 2 min pre-incubation with 4 ng/ml triethyltin to assess the oligomycin-insensitive ATP hydrolysis flux.

## BN-PAGE analyses and two-dimensional electrophoresis on total yeast cells protein extracts

For CN and BN-PAGE, 100 or 200 μg of total cell protein extracts were solubilized with Glyco-diosgenin (GDN) (0.5 g/g) or high-purity digitonin (1.5 g/g) in extraction buffer (see above) with 0.0125 kU/μl of nuclease. Membranes were solubilized by vortexing for 30 min at 4 °C and incubated at room temperature for 10 min. Supernatants were collected after centrifugation of the solubilized protein extract at $30,000 \times g$ for 30 min. The loading buffer containing 0.15 M 6-aminohexanoic acid was used for CN-PAGE and was supplemented with 20% (w/v) glycerol and 0.0125% (w/v) Coomassie brilliant blue G-250 for BN-PAGE. Proteins samples were loaded on Bis-Tris Invitrogen™ Novex™ NativePAGE™ 3–12% acrylamide gradient gels. Gel migration was performed at 10 mA, 3 h at 4 °C. At three-quarters of the migration, the BN-PAGE buffer was removed and replaced by a CN-PAGE buffer to decrease the blue coloration of gel.

Protein complexes were detected by in-gel activity as previously described (Molinié et al, 2022). Native gels were incubated with activity buffers containing 50 mM $KH_2PO_4$ pH 7.4 and 0.5 mg/ml iodonitrotetrazolium. The buffer was complemented with 400 μM NADH pH 7.0 to reveal NADH dehydrogenases. For cytochrome c reductase in-gel activity, native gels were incubated in the following buffer: 50 mM $KH_2PO_4$ pH 7.4, 75 mg/ml sucrose, 0.5 mg/ml 3,3'-Diaminobenzidine and 1 mg/ml cytochrome c. For ATP synthase in-gel activity, native gels were incubated in the following buffer: 50 mM glycine, 1.32 mM lead acetate, 0.1% (w/v) Triton X-100, and

supplemented with 5 mM MgSO$_4$ and 4 mM ATP pH 7.0 to start the reaction. After revelation of ATPase activity, native gels were incubated with 0.1% (v/v) HCl to remove lead precipitate before Coomassie staining (0.125% (w/v) Coomassie, 50% (v/v) ethanol, 10% (v/v) acetic acid). After 45 min, gels were destained with a destaining solution (25% (v/v) ethanol and 8% (v/v) acetic acid). After in-gel activity, Native gels were imaged using ImageQuant (Amersham) and the Optical Density was determined using FIJI analyzer.

For two-dimensional electrophoresis (2D-BN/SDS-PAGE), the first dimension BN-PAGE bands were excised and incubated for 15 min in denaturing and reducing buffer containing 1% SDS and 1% β-mercaptoethanol pH 6.4, and then incubated in a second buffer containing 1% SDS pH 6.4 for 15 min. Each lane was placed at the top of Bis-Tris Invitrogen™ Novex™ NuPAGE™ 4–12% acrylamide gradient gels and a gel solution (4% of acrylamide) was poured to seal the lane. PageRuler™ Plus (10 to 250 kDa) were loaded as a MW ladder. The migration was performed at 100 V for 1h30 in Novex™ MES running buffer according to the manufacturer recommendations.

## Quantitative Western blot analyses

For quantitative Western blot analyses, 50 or 100 μg of protein extracts were solubilized in standard RIPA buffer containing 150 mM NaCl, 25 mM Tris-base pH 8.0, 1% (w/v) NP40, 1% (w/v) SDS, 0.25% (w/v) deoxycholate and 1 mM EGTA for 30 min at 4 °C. The loading buffer containing 0.3 M Tris pH 6.8, 50% (w/v) glycerol, 30% (v/v) thioglycerol, 10% (w/v) SDS, 0.05% (w/v) bromophenol blue was added to the samples before incubating them at 75 °C for 5 min. To obtain a good separation of small proteins we performed a denaturing Schägger gel using 15% of acrylamide gels as previously described (Schägger and von Jagow, 1987). A molecular weight marker, PageRuler™ Plus (10 to 250 kDa), was loaded on the gel. The denaturing electrophoresis was performed at 100 V for 2 h. Proteins were then transferred onto nitrocellulose Amersham Protran Premium membrane (Amersham) (0.2 μm) with transfer buffer containing 25 mM Tris, 192 mM glycine, 0.1% (w/v) SDS at 100 V during 1 h at 4 °C. Membranes were stained with ponceau red solution containing 2 mg/ml ponceau red with 31% (v/v) acetic acid. After, membranes were incubated with blocking buffer containing 5% (w/v) skimmed milk diluted in PBS-tween buffer containing 10 mM NaH$_2$PO$_4$ pH 7.2, 0,14 M NaCl, 0.1% (w/v) Tween-20. For immunodetection, membranes were incubated with primary antibody diluted in PBS-tween and detected by a peroxidase-conjugated secondary antibody Clarity ECL reagent (Bio-Rad). The chemiluminescence signals were recorded using an ImageQuant (Amersham) and then quantified using ImageJ software.

## If1 and Stf1 proteins were produced and purified to determine the relative abundance of both inhibitors in cells

If1 was expressed and purified as mentioned in Corvest et al (2007), except that the His-tag was not removed. The *STF1* gene was amplified by PCR using the 5′-CGCGCGCCATGGCTGTTCTCATCATCAT CATCACGACGGTCCTCGTGTGTGTGTGCCGG-3′ forward primer to introduce a N-terminal His-tag and the 5′-CGCGCGCCATGGCT GTTCTCATCATCATCATCACGACGGTCCTCTCGTGTGTGTGTG TGCCGG-3′ reverse primer. The PCR product, digested by *Nco*1 and *Bam*H1, was inserted into the pIVEX2.3 vector and digested with the same restriction enzymes. The protein was produced in cell-free expression system (Larrieu et al, 2017). After 18 h of production at 28 °C, the reaction mix was centrifuged (10 min, 12,000 × *g*, 4 °C) and the supernatant containing Stf1 was diluted 10-fold and loaded onto a Nickel NTA column. The column was washed with 25 column volumes of washing buffer 1 containing 150 mM NaCl, 10 mM imidazole, 20 mM Tris-HCl pH 8.0, containing EDTA-free protease inhibitors (Pierce) and then with 20 column volumes of washing buffer 2 containing 150 mM NaCl, 20 mM imidazole, 20 mM Tris-HCl pH 8.0. The elution was performed with 4 column volumes of elution buffer containing 150 mM NaCl, 250 mM Imidazole, 20 mM Tris-HCl pH 8.0 and EDTA-free protease inhibitors. To determine the relative abundance of If1/Stf1, a mix containing the two purified peptides was generated and validated using Coomassie staining. This If1/Stf1 mix was used for Figs. 3B, 5A and EV1F.

## Antibodies

Primary antibodies: Pgk1 mouse monoclonal antibody 22C5D8 (CiteAb 459250), MTCO2 (Cox2) mouse monoclonal antibody (Thermo Fisher 12C4F12). Rabbit polyclonal antibodies raised against purified subunits β, γ, and subunit 4 were obtained in the laboratory. Rabbit polyclonal antibodies raised against the INDPRNPRFAKGGK peptide of subunit i were purchased from Neosystem. Anti-If1 antibodies were kindly provided by K. Tagawa (Osaka, Japan). The Stf1 protein produced in vitro was used by Covalab society to raise polyclonal rabbit antibodies.

The secondary antibodies used were: Peroxidase AffiniPure™ Goat anti-rabbit IgG (Jackson ImmunoResearch AB_2313567); Peroxidase AffiniPure™ Goat anti-mouse IgG (Jackson ImmunoResearch AB_2338504).

## Metabolites quantifications

Cells were cultured for 12 h before adding 1 μM CCCP at 1 OD$_{550nm}$. After 1 h of treatment, size and number of cells were defined with a multisizer instrument. 20 ml of culture were filtered (Sartolon polyamid 0.45 μm) and the filter was rapidly rinsed twice with ice-cold water to stop reactions. Metabolites were extracted using an ethanol/20 mM HEPES pH 7.2 (2/8 v/v) solution as described (Ceschin et al, 2014). Metabolites were separated, detected, and quantified on a High Performance Ion Chromatography (HPIC) station as described (Pinson et al, 2023). The intracellular concentration of nucleotides was determined using standard curves obtained with pure compounds. Adenylate energy charge was defined as AEC = [ATP] + ½ [ADP]/[ATP] + [ADP] + [AMP] (Atkinson and Walton, 1967).

## Mitochondrial DNA quantification

For DNA extraction, cells were washed and broken by shaking during 1 min with an equal volume of glass beads (0.4 mm) in the following buffer: 10 mM Tris pH 8.0, 1 mM EDTA, 100 mM NaCl, 2% (v/v) Triton X-100, 1% (v/v) SDS and 50% (v/v) chloroform/phenol (1:1). Equal volumes of 10 mM Tris pH 8.0 and 1 mM EDTA buffer were added and vortexed during 5 min. The supernatant was mixed with an equal volume of chloroform during

1 min and centrifuged at $12,000 \times g$. The supernatant was mixed with a double volume of glacial ethanol and centrifuged at $12,000 \times g$. The pellet was dried at room temperature before being resuspended in nuclease-free water.

Quantitative PCR for mtDNA content were performed with qPCRBIO SyGreen Blue Mix Lo-ROX (Eurobio®). Based on the manufacturer's instructions, 0.1 ng of DNA was used to quantify mtDNA and nuclear DNA with two different sets of primers. The first mtDNA primer set (fw: TTGAAGCTGTACAACCTACC, rv: CCTGCGATTAAGGCATGATG) targeted a region of *COX3* gene, the second primer set (fw: AACAATTGGTTTATTAGGAGCAGG-TATTGG, rv: TATACACCGAATAATAATAAGAATGAAACC) targeted a region of *ATP9* gene. The first nDNA primer set (fw: CACCCTGTTCTTTTGACTGA, rv: CATAGAAGGCTGGAACGT TG) targeted a region of *ACT1* gene, and the other primer set (fw: TGCTTTGTCAAATGGATCATATGG, rv: CCTGGAACCAAGT-GAACAGTAC) targeted a region of *GAL1* gene.

## Statistical analyses

Data are presented as mean ± SEM. Sample numbers (different culture) and experimental repeats are indicated in legends. Data were analyzed with the GraphPad Prism software using unpaired Student's t-test, one-way ANOVA or two-way ANOVA. A 0.05 *p*-value was considered statistically significant.

# Data availability

This study includes no data deposited in external repository.

The source data of this paper are collected in the following database record: biostudies:S-SCDT-10_1038-S44319-025-00430-8.

# Peer review information

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

## Acknowledgements

We thank Deborah Tribouillard-Tanvier's team, Muriel Priault's team, Anne Devin's team and Derek McCusker's team for sharing advice, instruments, consumables, and antibodies. Sabine Vaur and all members of the IBGC support team. We thank Dr Mairead Aubert for English proofreading of the manuscript. We thank also Claudine David for sharing precious advice and expertise and for her important support. The authors thank the staff of the SAM platform from TBMCore unit (University of Bordeaux - CNRS UAR 3427 - INSERM US05) for quantification of nucleotides. The HPIC chromatography station used for nucleotide determination was purchased with the financial support of both SIRIC BRIO (COMUCAN) and the Region Nouvelle-Aquitaine (MetabOptic 2022-24564910, AAPPF2021-2020-12000110). This work was supported by grants from ANR (DynaMitoPatho ANR-22-CE14-0040) and University of Bordeaux (SBM-AAPG-2024). We wish to honor the life and work of Isabelle Larrieu, whose exceptional professionalism, generous spirit, and kindness are deeply missed.

## Author contributions

**Orane Lerouley**: Conceptualization; Data curation; Formal analysis; Investigation; Methodology. **Isabelle Larrieu**: Resources; Investigation. **Tom Louis Ducrocq**: Resources; Investigation. **Benoît Pinson**: Resources; Funding acquisition; Investigation; Methodology. **Marie-France Giraud**: Conceptualization; Resources; Supervision; Funding acquisition; Investigation. **Arnaud Mourier**: Conceptualization; Resources; Data curation; Formal analysis;

Supervision; Funding acquisition; Investigation; Methodology; Writing—original draft; Project administration; Writing—review and editing.

Source data underlying figure panels in this paper may have individual authorship assigned. Where available, figure panel/source data authorship is listed in the following database record: biostudies:S-SCDT-10_1038-S44319-025-00430-8.

## Funding

## Disclosure and competing interests statement

The authors declare no competing interests.

# Expanded View Figures

**Figure EV1. If1/Stf1 are required to maintain the $F_1F_0$-ATP synthase-free $F_1$ subcomplex level and activity.**

BN-PAGE (3–12%) (**A**) and densitometric analysis (**B**) performed with total cell extracts, from WT, *inh1Δ stf1Δ*, and *atp18Δ* grown under glycerol 2% rich medium. Cell extracts were solubilized with increasing digitonin-to-protein ratio ranging from 0.5 to 6 g/g protein. The ATP synthase assemblies (O: oligomers; V: monomers; $F_1$: free $F_1$ subcomplex) were revealed by $F_1F_0$-ATP synthase (CV) hydrolytic in-gel activity (IGA). ($n = 3$, from left to right $*p = 0.0171$, $**p = 0.0081$, $**p = 0.0046$, $*p = 0.0441$, $**p = 0.0045$, $***p = 0.0007$, two-way ANOVA, error bars ± SEM). (**C**) Measurement of the ATP hydrolysis flux performed on purified mitochondria from total cell extracts from WT (black bars) and *inh1Δ* (red bars) grown on lactate 2% rich medium by monitoring the ATP induced phosphate production flux over several minutes. Experiments were performed at pH 9.0 (inactive inhibitors) and pH 6.4 (active inhibitors). ($n \geq 7$ independent experiments, $***p < 0.0001$, unpaired t-test, error bars ± SEM). (**D**) Determination of doubling time (hours) of WT (black bars) and *inh1Δ stf1Δ* (red bars) grown on different fermentable (glucose 0.5%, galactose 2%) and non-fermentable (glycerol 2%, lactate 2%) culture-rich media supplemented or not with CCCP (1, 1.5 or 2 μM), following the optical density of the culture at 550 nm. *nd*: too slow to determine the doubling time ($n = 3$ independent experiments, $**p = 0.0017$, two-way ANOVA, error bars ± SEM). (**E**) BN-PAGE (3–12%) performed with total cell extracts from WT grown on different fermentable (glucose 0.5%, galactose 2%) and non-fermentable (glycerol 2%, lactate 2%) culture-rich media. Cell extracts were solubilized with digitonin at a digitonin-to-protein ratio of 1.5 g/g protein. 150 μg of protein were loaded for glucose or galactose condition and 100 μg of protein were loaded for glycerol or lactate conditions. The $F_1F_0$-ATP synthase assemblies were revealed by $F_1F_0$-ATP synthase (CV) hydrolytic in-gel activity (IGA). (Representative of $n = 2$ independent experiments). (**F**) Western blot (left) and densitometric analysis (right) of the relative abundance of If1 and Stf1, using Pgk1 as loading control. Denaturing electrophoresis was performed with total cell extracts grown on glycerol 2% rich medium supplemented or not with CCCP. The Coomassie blue staining on the upper panel demonstrates the equal loading of tagged If1 and Stf1 produced in vitro and used for the relative quantification of inhibitors (* signal remaining from anti-STF1 antibody). The Coomassie staining of the purified peptides is also presented in Figs. 3B and 5A. ($n = 3$ independent experiments, unpaired t-test, error bars ± SEM). (**G**) Densitometric analysis of drop test performed on WT (black) and *atp18Δ* (orange) mutant grown on glycerol 2% culture minimum medium. ($n = 3$ independent experiments, from left to right $***p < 0.0003$, $***p < 0.0004$, $*p = 0.0227$, two-way ANOVA, error bars ± SEM). (**H**) Densitometric analysis of drop test performed on WT (black) and *cdc19Δ* (green) thermosensitive (ts) mutant (37°) mutant grown on glycerol 2% culture minimum medium. ($n = 3$ independent experiments, from left to right $***p < 0.0001$, $***p = 0.0003$, two-way ANOVA, error bars ± SEM). (**I**) Densitometric analysis of drop test performed on WT (black) and *cdc19Δ* (green) thermosensitive (ts) mutant (37°) mutant grown on lactate 2% culture minimum medium. ($n = 3$ independent experiments, from left to right $***p < 0.0004$, $***p < 0.0001$, $**p = 0.0088$, two-way ANOVA, error bars ± SEM).

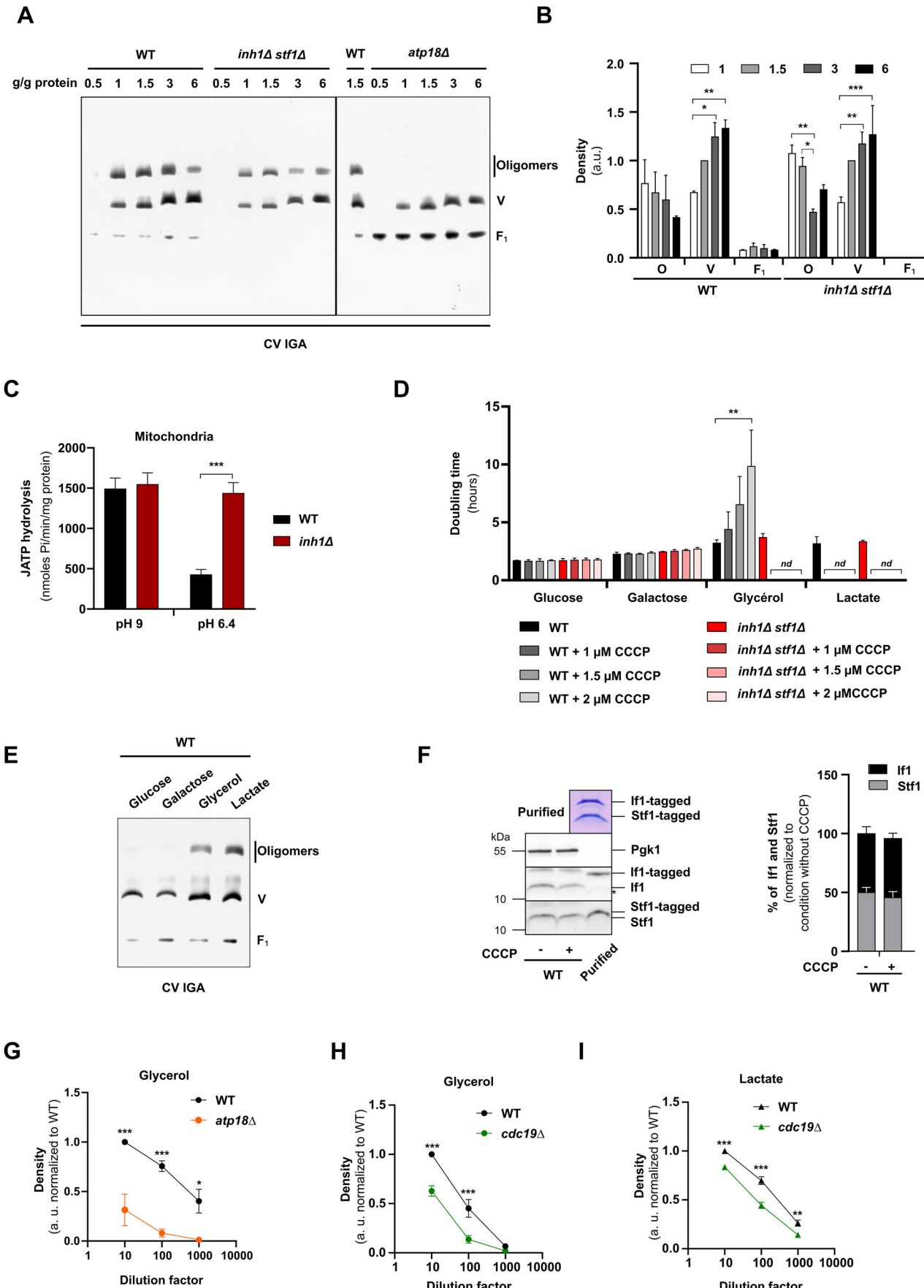

## A

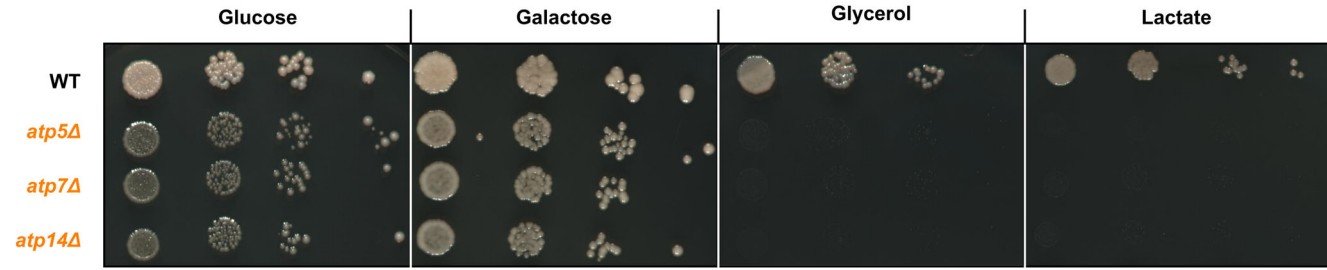

**Figure EV2. The growth phenotype of F₁F₀-ATP synthase-deficient strains on various carbon sources.**

(A) Drop test performed on WT, atp5Δ, atp7Δ, and atp14Δ mutant grown on different fermentable (glucose 0.5%, galactose 2%) or non-fermentable (glycerol 2%, lactate 2%) culture minimum media. (Representative of $n = 3$ independent experiments).

