## [Peer Review File · EMBO Reports]

An alternative mechanism by which If1 prevents ATP hydrolysis by the ATP synthase subcomplex in *S. cerevisiae*

Orane Lerouley, Isabelle Larrieu, Tom Ducrocq, Benoit Pinson, Marie-France Giraud, and Arnaud Mourier

Corresponding author(s): Arnaud Mourier (arnaud.mourier@ibgc.cnrs.fr)

Review Timeline:

Transfer Date:	12th Dec 24
Editorial Decision:	24th Jan 25
Revision Received:	31st Jan 25
Accepted:	6th Mar 25

Editor: Deniz Senyilmaz Tiebe

Transaction Report: This manuscript was transferred to EMBO reports following peer review at Review Commons.

**Review
COMMONS**

Review #1

1. Evidence, reproducibility and clarity:

Evidence, reproducibility and clarity (Required)

Here the authors investigate the impact of the presence of the ATP synthase-specific inhibitory proteins IF1 and STF1 on the assembly and function of yeast F1Fo under various metabolic conditions. They make the important discovery that when the genes for the two inhibitors are knocked out, that the amount of soluble F1 present becomes undetectable. This is a significant contribution to understanding the roles of IF1 and STF1 in the assembly of F1Fo and in the metabolic regulation of the cell. The evidence is strong, the results are reproducible and clear.

There are no major concerns about the results or their interpretation. However, the English grammar needs a great deal of work, and it is recommended that a native English speaking scientist help in the revision. There are too many issues with the grammar (e.g. singular/plural) to describe. The authors must replace the word 'comfort', which appears often in the discussion with a word that more accurately describes what they intend to say.

2. Significance:

Significance (Required)

This work provides the first evidence that soluble F1 is present in significant amounts during the normal growth of yeast and is not an artifact of cellular disruption. The results are significant in that they provide a new understanding of the role of IF1 and STF1 in the metabolic regulation of ATP synthesis and in the assembly of F1Fo. The last sentence of the discussion makes the point that these results have important implications regarding cellular energy balance in the growth of tumors.

3. How much time do you estimate the authors will need to complete the suggested revisions:

Estimated time to Complete Revisions (Required)

(Decision Recommendation)

Less than 1 month

4. Review Commons values the work of reviewers and encourages them to get credit for their work. Select 'Yes' below to register your reviewing activity at Web of Science Reviewer Recognition Service (formerly Publons); note that the content of your review will not be visible on Web of Science.

Yes

Review #2

1. Evidence, reproducibility and clarity:

Evidence, reproducibility and clarity (Required)

****Summary:****

Provide a short summary of the findings and key conclusions (including methodology and model system(s) where appropriate).

The present manuscript, authored by Orane Lerouley et al., focuses on the role of inhibitory factor 1 (IF1) in *Saccharomyces cerevisiae*. It is established that IF1 has the specific function of inhibiting the reverse ATP hydrolytic activity of ATP synthase. To challenge some of the established concepts, the team employed a range of complementary methods, culture conditions and yeast strains. The results demonstrate that IF1/STF1 are not essential for the stability of F1-F₀ ATP synthase oligomers. However, they are crucial for maintaining the levels of free F1 subcomplexes. Therefore, the free F1 subcomplexes observed in the control strain are not a degradation by-product resulting from digitonin treatment, as previously documented. Instead, they are free F1 stabilized in the mitochondrial matrix by IF1, which prevents toxic and futile ATP hydrolysis. Additionally, they discovered that free F1 is not a prerequisite for cells to withstand mtDNA loss, as previously documented. Furthermore, they demonstrated that the glycerol carbon source, unlike lactate, is not a strictly "non-fermentable" carbon source that depends exclusively on mitochondrial energy metabolism. Instead, it also exploits the glycolytic pathway.

In conclusion, their data demonstrated that preventing mitochondrial ATP hydrolysis is critical for cell growth only when energy metabolism relies equally on both glycolysis and OXPHOS processes. The manuscript is well organised and the aim of the study is clearly stated in the introduction. The results are robust and clearly presented.

****Major comments:****

- Are the key conclusions convincing?

The key conclusions are robust and convincing. The authors exploited a variety of yeast strains, culture conditions with different carbon sources, and a range of methods, including cell growth in liquid and solid media, SDS-PAGE, WB, BN-PAGE, CN-PAGE, in-gel activities, measurement of ATP hydrolysis, oxygen consumption, adenyl energy charge, mtDNA quantification, and rho+/rho-percentage.

- Should the authors qualify some of their claims as preliminary or speculative, or remove them altogether?

The claims are all well-founded and supported by evidence.

With regard to prospects, the authors suggest that the findings may help to shed light on the controversial role of IF1 in cancer. It appears that the mechanisms are different in yeast and humans. Further investigation in human cells would have helped to reach this conclusion.

- Would additional experiments be essential to support the claims of the paper? Request additional

experiments only where necessary for the paper as it is, and do not ask authors to open new lines of experimentation.

In Figure 1, single inh1 and stf1 mutants would provide controls and additional information.

In Figure 3D, triple inh1 stf1 atp18 mutant should be added to be sure of the conclusion « Intriguingly, the combined loss of IF1 and free F1 subcomplex in this atp18 Δ strain did not worsen nor rescue its growth deficiency » since two homologous inhibitory peptides, namely IF1 and STF1 have been identified in *S. cerevisiae*. Double inh1 stf1 and single inh1 mutants are also missing.

In Figure 5C, the use of two different mutants for each pathway (glycolytic and oxidative) would have allowed more reliable conclusions to be drawn. See the paper by Galkina et al, *Frontiers in microbiology*, 2022, where glycerol is used as a non-fermentable substrate.

- Are the suggested experiments realistic in terms of time and resources? It would help if you could add an estimated cost and time investment for substantial experiments.

The suggested experiments are all realistic in terms of time and resources. The estimated cost is a few commercial strains and a few PCRs, and the time investment is no more than 1 month.

Furthermore, it is very likely that the authors already have the strains and tools.

- Are the data and the methods presented in such a way that they can be reproduced?

Data and methods are easily reproducible.

- Are the experiments adequately replicated and statistical analysis adequate?

Experiments are adequately replicated and statistically well analyzed.

****Minor comments:****

- Specific experimental issues that are easily addressable.

In Figures 1B, 3D, 4B, and 6C, means and standard deviations should be shown on the graphs instead of a representative curve.

- Are prior studies referenced appropriately?

Authors should add the following publications :

in line 92 : "The inhibitor protein IF1 from mammalian mitochondria inhibits ATP hydrolysis but not ATP synthesis by the ATP synthase complex" by Joe Carroll, Ian N. Watt, Charlotte J. Wright, Shujing Ding, Ian M. Fearnley , and John E. Walker, *JBC*, 2024.

« Heterogeneity of Starved Yeast Cells in IF1 Levels Suggests the Role of This Protein in vivo » by Kseniia V. Galkina, Valeria M. Zubareva, Nataliia D. Kashko, Anna S. Lapashina, Olga V. Markova, Boris A. Feniouk and Dmitry A. Knorre, *Frontiers in microbiology*, 2022 to document the role of IF1 and to discuss the glyco-oxidative properties of glycerol.

- Are the text and figures clear and accurate?

Authors need to be a little more consistent in their writing. For example, all Latin words should be italicized, the 1 in IF1 and F1 should be subscripted, and the su of subunit should always be Su or su.

All reagents must be referenced.

2. Significance:

Significance (Required)

Provide contextual information to readers (editors and researchers) about the novelty of the study, its value for the field and the communities that might be interested.

****The following aspects are important:****

- ***General assessment:** provide a summary of the strengths and limitations of the study. What are the strongest and most important aspects? What aspects of the study should be improved or could be developed?

The strength of the study is the use of the powerful model organism yeast *Saccharomyces cerevisiae*, which allows different conditions to be easily tested in several different strains of interest. Furthermore, the multiplicity of approaches allows to be more confident with the conclusions. The limitation of the study is the exclusive use of yeast. The strongest and most important aspects are the challenging of existing dogmas. The aspects of the study that should be improved or could be developed are the role of IF1/STF1 in different stress conditions (in addition to the CCCP stress condition) and the conclusion about the glycerol carbohydrate source. They stated that glycerol is a glyco-oxidative metabolic condition, but more experiments need to be done before concluding.

- ***Advance:** compare the study to the closest related results in the literature or highlight results reported for the first time to your knowledge; does the study extend the knowledge in the field and in which way? Describe the nature of the advance and the resulting insights (for example: conceptual, technical, clinical, mechanistic, functional,...).

To my knowledge, this is the first time that IF1/STF1 has been shown to be important for the stabilization of free F1 subcomplexes and not for the stability of the F1-F0 ATP synthase oligomers. It is also the first time that free F1 is shown not to be required for cells to survive to mtDNA loss. Finally, it is the first time that glycerol is shown to be glyco-oxidative and not strictly a "non-fermentable" carbon source that depends exclusively on mitochondrial energy metabolism.

This study provides further insight into the physiological and structural role of the IF1 inhibitor. It also provides mechanistic insights and may open doors to clinical research by better understanding the role of IF1, whose expression is deregulated in many diseases.

- ***Audience:** describe the type of audience ("specialized", "broad", "basic research", "translational/clinical", etc...) that will be interested or influenced by this research; how will this

research be used by others; will it be of interest beyond the specific field?

The audience interested in this research may be "specialized" in IF1 or ATP synthase, "broad" using yeast as a model, "translational/clinical" working on diseases involving IF1.

- Please define your field of expertise with a few keywords to help the authors contextualize your point of view. Indicate if there are any parts of the paper that you do not have sufficient expertise to evaluate.

My field of expertise is mitochondrial diseases

3. How much time do you estimate the authors will need to complete the suggested revisions:

Estimated time to Complete Revisions (Required)

(Decision Recommendation)

Less than 1 month

4. Review Commons values the work of reviewers and encourages them to get credit for their work. Select 'Yes' below to register your reviewing activity at Web of Science Reviewer Recognition Service (formerly Publons); note that the content of your review will not be visible on Web of Science.

No

Review #3

1. Evidence, reproducibility and clarity:

Evidence, reproducibility and clarity (Required)

Lerouley et al describe a series of concise, well executed experiments to understand the role of the inhibitory factors IF1/STF1 in yeast metabolism. They discover that IF1/STF1 are important in maintaining the stability of soluble F1 subcomplexes and that they have an important metabolic role in maintaining cell growth under 'glyco-oxidative' conditions, where the carbon source (glycerol) can be used to generate ATP via both glycolysis and oxidative phosphorylation.

All conclusions and claims are well supported. My main and only complaint is the quality of the sentence construction which often made it difficult to understand the intentions of the authors.

Below is a list of issues which need to be resolved prior to acceptance. Most are related to sentence construction, consistent nomenclature and use of 'unusual' expressions.

****Comments relating to contents****

Line 155/Figure S1A - please can you provide a densitometry graph specifically showing the changes in free F1 subcomplexes to oligomers. I'm not seeing what the authors are describing.

Line 288/figure 5C - There are two panels with WT grown on Glycerol. The lower panel shows poor growth of WT on glycerol and the amount of growth is similar to *pyk1* delta mutants. Thus how do you validate the conclusion that the WT and *pyk1* delta grow differently.

Line 306: Loss of IF1/STF1 causes loss of F1 subcomplex stability. How does that influence the observation of the growth difference on glycerol

DOI: 10.1073/pnas.1816556116 show that F-type ATP synthase dimers can form rows and induce membrane curvature just from the intrinsic shape of the dimer without the need for IF1 connecting neighbouring dimers together. This was supported by coarse grain molecular dynamics simulation DOI: 10.1085/jgp.201812033, DOI: 10.1073/pnas.1204593109. The findings of these papers should be included in the discussion.

****Comments relating to sentence construction/word usage****

Line 23: ...living world.

Line 42: Nevertheless, in cells presenting defective mitochondrial genome levels... (no 's' on defective)

Line 55: ...generating a proton electrochemical potential...

Line 165: factors not actors

Line 168: second dimensional gel (Figure 2A). (including word gel will help with clarity of sentence)

Line 171: inclusion of the word drastic might be too dramatic.

Line 179: replace Then with Next.

Line 218: include name of mutant to improve clarity of sentence. E.g. The CN and BN-PAGE performed on the solubilized ATP synthase from total protein cell extract 'of the delta18 mutant'

Also sentence needs to be shortened or divided into multiple sentences.

Line 227: when you say F1F0-ATP synthase are you talking about monomer or oligomer or even F1

subcomplex? This needs to be clarified.

Line 237 and 238. Replace and with or in the brackets. The use of and suggests the media has both glucose and galactose in.

Line 250: I don't understand what is being 'gradually increased' or how.

Line 258: remove 's from respiration's responses

Line 286: I don't understand the phrase 'energy metabolic'

Figure 5C: coloured bar at bottom should be oxidative (not oxydative).

Line 359: I don't think 'structuration' is a word. 'Organization' would be more appropriate

Line 361: I think comforted should be confirmed. This also occurs on line 393 and line 395.

Line 492: remove last 'e' from supercomplexe.

Line 557: 'assessed extemporaneously'. I have no idea what that means or whether it is a word.

Line 605: I don't find the wording clear here. What do you mean by 'spontaneously partially'

There is supplementary figures S1, S3 and S5 but no S2 or S4.

Methods:

General comment about buffer conditions. Please use one format. In the methods and text you have the following varieties:

20mM chemical
chemical 20mM
chemical (20mM)

This ends up being confusing as to which value belongs to which chemical. It also means some chemicals end up with no value. E.g. methionine or uracil doesn't have a quantity (line 690)

Line 672: I think respectively needs to be at end of sentence.

Line 674: 'with glucose' should be 'containing glucose'

Line 676: The P-/o colonies were identified and counted by comparing to a replica plate containing glycerol...

Line 690: missing value for either methionine or uracil.

Line 694: 'thermostated at 28oC'. No need for 'thermostated'

Line 698: concentration of CCCP in brackets should be moved to after titration in line 697
Line 722: remove 'concentrated with'
Line 746: replace experimentation with experiment.
Line 791. The sentence starting with 'The Novex MES' doesn't make sense (missing a verb).
Line 802: electrophoresis
Line 824: Please define CV. During manuscript CV was ATP synthase. I presume here it means column volume.
Line 850 & 854: I think for is more appropriate that 'during'
Line 858 and 859: 'was performed' and 'was used'.
Line 885: remove 's' from advices.

****Referees cross-commenting****

No additional comments but I agree with Reviewer 1 that the paper needs to be read and edited by a native or expert English speaker.

2. Significance:

Significance (Required)

Very short, concise and clear piece of work addressing an important and overlook aspect of bioenergetics. It will have significant impact on the field specifically related to the clear role they have uncovered for IF1/STF1 and the importance of investigating bioenergetic processes in relation to carbon source and oxphos vs glycolysis. Hopefully this work will help eradicate the false concept that IF1/STF1 are important in ATP synthase oligomer formation. This is clearly disproved in this manuscript.

It is basic science but has a significant impact on the medical and bioenergetics field.

3. How much time do you estimate the authors will need to complete the suggested revisions:

Estimated time to Complete Revisions (Required)

(Decision Recommendation)

Less than 1 month

4. Review Commons values the work of reviewers and encourages them to get credit for their work. Select 'Yes' below to register your reviewing activity at Web of Science Reviewer Recognition Service (formerly Publons); note that the content of your review will not be visible on Web of Science.

Yes

Full Revision

Manuscript number: RC-2024-02639

Corresponding author(s): Arnaud Mourier

1. General Statements

Dear Dr. David del Alamo, we thank you for providing us with expert reviewers and we appreciate their very positive and constructive comments:

Reviewer 1: «They make the important discovery that when the genes for the two inhibitors are knocked out, that the amount of soluble F1 present becomes undetectable. This is a significant contribution to understanding the roles of IF1 and STF1 in the assembly of F1Fo and in the metabolic regulation of the cell. The evidence is strong, the results are reproducible and clear...The results are significant in that they provide a new understanding of the role of IF1 and STF1 in the metabolic regulation of ATP synthesis and in the assembly of F1Fo. »

Reviewer 2: « The manuscript is well organised and the aim of the study is clearly stated in the introduction. The results are robust and clearly presented...The key conclusions are robust and convincing....The claims are all well-founded and supported by evidence. »

Reviewer 3: « Lerouley et al describe a series of concise, well executed experiments to understand the role of the inhibitory factors IF1/STF1 in yeast metabolism. All conclusions and claims are well supported...Very short, concise and clear piece of work addressing an important and overlook aspect of bioenergetics. It will have significant impact on the field specifically related to the clear role they have uncovered for IF1/STF1 and the importance of investigating bioenergetic processes in relation to carbon source and oxphos vs glycolysis. Hopefully this work will help eradicate the false concept that IF1/STF1 are important in ATP synthase oligomer formation. This is clearly disproved in this manuscript...It is basic science but has a significant impact on the medical and bioenergetics field. »

We appreciate the time and effort the reviewers have dedicated to evaluating our work. We have carefully considered each of the reviewers' suggestions and have revised the manuscript accordingly. Below, we provide a point-by-point response to the reviewers' comments.

This section is mandatory. Please insert a point-by-point reply describing the revisions that were already carried out and included in the transferred manuscript.

Reviewer #1:

Comment: "The English grammar needs a great deal of work, and it is recommended that a native English speaking scientist help in the revision. There are too many issues with the grammar (e.g. singular/plural) to describe. The authors must replace the word 'comfort', which appears often in the discussion with a word that more accurately describes what they intend to say."

Full Revision

The manuscript has been carefully proofread by Dr. Mairead Aubert, a professional native English speaker, who has made significant corrections to the grammar and phrasing, greatly improving the overall quality of the article.

Reviewer #2:

Comment: “In Figure 1, single *inh1* and *stf1* mutants would provide controls and additional information.”

*In agreement with reviewer 1, we have performed additional experiments. First, we have assessed the growth of single and double mutant strains on Lactate 2% medium (Figure 1B). Then, total cell extracts solubilized with digitonin from WT, *inh1*Δ *stf1*Δ, *inh1*Δ and *stf1*Δ, grown on glycerol 2% were subjected to BN-PAGE analyses (Figure 1F). In line with reviewer 1 comment, these added data further strengthen our conclusion showing that (i) the levels of ATP synthase monomers and higher F_1F_0 -ATP synthase oligomers were not impacted by the individual or combined loss of both *If1* and *Stf1*, (ii) that the F_1 subcomplex level was more severely reduced in *inh1*Δ than in *stf1*Δ, and was hardly detected in *inh1*Δ *stf1*Δ samples (Figure 1D-F).*

Comment: “In Figure 3D, triple *inh1 stf1 atp18* mutant should be added to be sure of the conclusion « Intriguingly, the combined loss of IF1 and free F1 subcomplex in this *atp18*Δ strain did not worsen nor rescue its growth deficiency » since two homologous inhibitory peptides, namely IF1 and STF1 have been identified in *S. cerevisiae*. Double *inh1 stf1* and single *inh1* mutants are also missing.”

*Following reviewer 2's request, we have engineered and characterized the double *atp18*Δ *stf1*Δ and triple *atp18*Δ *inh1*Δ *stf1*Δ mutant strains. Drop tests and BN-PAGE analyses were conducted to characterize (i) the growth of the different mutant strains on Glycerol 2% (Figure 3D,E), and (ii) ATP synthase assembly (Figure 3F). Our findings reveal that the combined loss of the inhibitory peptides (*inh1*Δ *stf1*Δ), which further reduces the level of the free F_1 subcomplex compared to *inh1*Δ, severely hampers the growth of *atp18*Δ mutant on glycerol (Figure 3D,E). These results confirm the cooperative activity of *If1* and *Stf1*, and unravel that both inhibitory peptides are essential to preserve the growth of *atp18*Δ mutant on glycerol.*

In Figure 5C, the use of two different mutants for each pathway (glycolytic and oxidative) would have allowed more reliable conclusions to be drawn. See the paper by Galkina et al, *Frontiers in microbiology*, 2022, where glycerol is used as a non-fermentable substrate.

We acknowledge this limitation and have included 3 mutants *atp5*Δ, *atp7*Δ, *atp14*Δ presenting an abolished OXPHOS-driven ATP synthesis, deleting ATP synthase subunits (Figure S5D). In line with previous reports, we observed that the abolished OXPHOS-driven ATP synthesis, prevent the capacity of these mutant strain to grow under oxidative (Lactate 2%) and Glyco-oxidative (Glycerol 2%) conditions. On the other hand, we could not find any other yeast mutants affecting the downstream part of glycolysis that would provide additional support to the

data obtained with the thermosensitive *cdc19Δ* mutant. The data obtained with the ATP synthase mutants are discussed and presented in Figure S5D.

Comment: In Figures 1B, 3D, 4B, and 6C, means and standard deviations should be shown on the graphs instead of a representative curve.

We agree and now provide in Supplemental Figure 4A a quantitative and statistical analyses comparing the doubling time of WT and *inh1Δ stf1Δ* strains on different carbon sources and in presence of various concentrations of CCCP.

Comment : Authors should add the following publications : in line 92 : "The inhibitor protein IF1 from mammalian mitochondria inhibits ATP hydrolysis but not ATP synthesis by the ATP synthase complex" by Joe Carroll, Ian N. Watt, Charlotte J. Wright, Shujing Ding, Ian M. Fearnley , and John E. Walker, JBC, 2024. « Heterogeneity of Starved Yeast Cells in IF1 Levels Suggests the Role of This Protein in vivo » by Kseniia V. Galkina, Valeria M. Zubareva, Nataliia D. Kashko, Anna S. Lapashina, Olga V. Markova, Boris A. Feniouk and Dmitry A. Knorre, Frontiers in microbiology, 2022 to document the role of IF1 and to discuss the glyco-oxidative properties of glycerol.

Thank you for this suggestion. We have included these citations.

Comment : Authors need to be a little more consistent in their writing. For example, all Latin words should be italicized, the 1 in IF1 and F1 should be subscripted, and the su of subunit should always be Su or su. All reagents must be referenced.

Thank you for highlighting these issues, we have corrected and abbreviation are now consistent across the manuscript.

Full Revision

Reviewer #3:

Comment : Line 155/Figure S1A - please can you provide a densitometry graph specifically showing the changes in free F₁ subcomplexes to oligomers. I'm not seeing what the authors are describing.

Following Reviewer 3 suggestion, we provide a densitometric analysis confirming that the digitonin titration strongly imbalance the levels of oligomers and monomers but does not modify the levels of free F₁ subcomplex.

Comment : Line288/figure 5C - There are two panels with WT grown on Glycerol. The lower panel shows poor growth of WT on glycerol and the amount of growth is similar to *pyk1delta* mutants. Thus how do you validate the conclusion that the WT and *pyk1delta* grow differently.

We agree with the reviewer that the growth of the WT strain at 37°C on glycerol and lactate are clearly affected. However, we now provide densitometric analysis performed on independent drop tests and confirmed that the growth of *cdc19Δ* is significantly affected compared to the WT in these conditions (Figure S5D, E). These additional analyses *strengthen our conclusions showing that the growth of the *cdc19Δ* is statistically and severely affected compared to the WT.*

Comment : Line 306: Loss of IF1/STF1 causes loss of F₁ subcomplex stability. How does that influence the observation of the growth difference on glycerol

Our results clearly indicate that under physiological condition the loss of If1/Stf1 associated with a severe loss of the free F₁ subcomplex does not significantly affect the growth on glycerol 2% (Figure 4). In contrast, the loss of If1/Stf1 is detrimental to the growth on glycerol under depolarizing stress conditions (CCCP, Figure 4) or in *atp18Δ* *presenting defective assembly and activity of the ATP synthase (Figure 3E)*. *Nevertheless, although glyco-oxidative conditions (2% glycerol) allow us to observe growth defects resulting from the loss of the If1/Stf1 inhibitory peptides, they do not clarify whether this stress is caused by the direct action of If1/Stf1 on the assembled ATP synthase, the loss of the F₁ subcomplex, or both.*

DOI: 10.1073/pnas.1816556116 show that F-type ATP synthase dimers can form rows and induce membrane curvature just from the intrinsic shape of the dimer without the need for IF1 connecting neighbouring dimers together. This was supported by coarse grain molecular dynamics simulation DOI: 10.1085/jgp.201812033, DOI: 10.1073/pnas.1204593109. The findings of these papers should be included in the discussion.

We appreciate this suggestion and have expanded the Discussion section to include and discuss these important findings.

Comments relating to sentence construction/word usage :

Line 23: ...living world.

Line 42: Nevertheless, in cells presenting defective mitochondrial genome levels... (no 's' on defective)

Full Revision

Line 55: ...generating a proton electrochemical potential...

Line 165: factors not actors

Line 168: second dimensional gel (Figure 2A). (including word gel will help with clarity of sentence)

Line 171: inclusion of the word drastic might be too dramatic.

Line 179: replace Then with Next.

Line 218: include name of mutant to improve clarity of sentence. E.g. The CN and BN-PAGE performed on the solubilized ATP synthase from total protein cell extract 'of the delta18 mutant'

....

Also sentence needs to be shortened or divided into multiple sentences.

Line 227: when you say F1F0-ATP synthase are you talking about monomer or oligomer or even F1 subcomplex? This needs to be clarified.

Line 237 and 238. Replace and with or in the brackets. The use of and suggests the media has both glucose and galactose in.

Line 250: I don't understand what is being 'gradually increased' or how.

Line 258: remove 's from respiration's responses

Line 286: I don't understand the phrase 'energy metabolic'

Figure 5C: coloured bar at bottom should be oxidative (not oxydative).

Line 359: I don't think 'structuration' is a word. 'Organization' would be more appropriate

Line 361: I think comforted should be confirmed. This also occurs on line 393 and line 395.

Line 492: remove last 'e' from supercomplexe.

Line 557: 'assessed extemporaneously'. I have no idea what that means or whether it is a word.

Line 605: I don't find the wording clear here. What do you mean by 'spontaneously partially'

There is supplementary figures S1, S3 and S5 but no S2 or S4.

Methods:

General comment about buffer conditions. Please use one format. In the methods and text you have the following varieties:

20mM chemical

chemical 20mM

chemical (20mM)

This ends up being confusing as to which value belongs to which chemical. It also means some chemicals end up with no value. E.g. methionine or uracil doesn't have a quantity (line 690)

Line 672: I think respectively needs to be at end of sentence.

Line 674: 'with glucose' should be 'containing glucose'

Line 676: The P-/o colonies were identified and counted by comparing to a replica plate containing glycerol...

Line 690: missing value for either methionine or uracil.

Line 694: 'thermostated at 28oC'. No need for 'thermostated'

Line 698: concentration of CCCP in brackets should be moved to after titration in line 697

Line 722: remove 'concentrated with'

Line 746: replace experimentation with experiment.

Line 791. The sentence starting with 'The Novex MES' doesn't make sense (missing a verb).

Full Revision

Line 802: electrophoresis

Line 824: Please define CV. During manuscript CV was ATP synthase. I presume here it means column volume.

Line 850 & 854: I think for is more appropriate that 'during'

Line 858 and 859: 'was performed' and 'was used'.

Line 885: remove 's' from advices.

We sincerely appreciate the time and effort spent by Reviewer 3 in identifying these errors and helping us enhance both the clarity and quality of our manuscript. We have addressed all of the suggestions and corrections, and the manuscript has been meticulously proofread by Dr. Mairead Aubert, a professional native English speaker, who has made significant improvements to the grammar and phrasing, thereby greatly enhancing the overall quality of the article.

We are grateful for the reviewers' thoughtful and constructive feedback, which has led to considerable improvements in our manuscript. We hope that the revised manuscript now meets the reviewers' expectations and are looking forward to your feedback.

Thank you once again for your valuable consideration.

Dear Dr. Mourier,

Thank you for submitting your revised manuscript, which was previously peer reviewed at Review Commons. It has now been seen by one of the original referees.

As you can see, the referee finds that the study is significantly improved during revision and recommend publication in EMBO Reports. However, I need you to address the points below before I can accept the manuscript.

- Please address the minor concerns of referee #3.
- Please provide 3-5 keywords for your study. These will be visible in the html version of the paper and on PubMed and will help increase the discoverability of your work.
- As per our guidelines, please add a 'Data Availability Section', where datasets and computer code that were generated in the reported study should be listed in a structured manner and placed after the Methods section. If your study does not include datasets, please insert the following statement: This study includes no data deposited in external repositories (please see <https://www.embopress.org/page/journal/14693178/authorguide#dataavailability> for further information).
- Please add a Disclosure Statement & Competing Interests section to the manuscript (<https://www.embopress.org/page/journal/14693178/authorguide#conflictofinterest>)
- Please remove the Author Contributions section from the manuscript text.
- As per our format requirements, in the reference list, citations should be listed in alphabetical order and then chronologically, with the authors' surnames and initials inverted; where there are more than 10 authors on a paper, 10 will be listed, followed by 'et al.'. Please see <https://www.embopress.org/page/journal/14693178/authorguide#referencesformat>
- Please fill out and include an author checklist as listed in our online guidelines (<https://www.embopress.org/page/journal/14693178/authorguide>)
- We note that the funding information regarding University of Bordeaux is currently missing from the manuscript tracking system.
- We note the following regarding the figure files:
 - o The main and supplementary figures provided in one PDF. Main figures need to be uploaded as separate production quality Figure files.
 - o The correct nomenclature for supplemental figures is Expanded View figures and their callouts are Figure EV1 etc. EV Figures also need to be provided as separate production quality Figure files.
 - o Suppl. figure 2 and its legend are currently missing, and it is not called out in the text.
 - o The figure PDF is not properly displayed in the Merged PDF - panel 1F is missing, Suppl. figure 2 is missing, panels 3EFG are missing, etc.
- All research articles submitted as revised versions must include a structured methods section that includes a Reagents and Tools Table followed by a Methods and Protocols section. This file should be submitted separately. Please see <https://www.embopress.org/page/journal/14693178/authorguide#structuredmethods> for further information.
- Please submit source data along with the revised manuscript as requested by our Source Data Coordinator Dr. Hannah Sonntag per email on 20.01.2025.
- The manuscript sections should be in the following order: Title page - Abstract & Keywords - Introduction - Results - Discussion - Methods - Data Availability - Acknowledgments - Disclosure Statement & Competing Interests - References - Figure Legends - (Main Tables with legends if applicable) - Expanded View Figure Legends.
- "Mat & meth" should be renamed as "Methods".
- During our routine figure checks, we note the following potential panel re-uses, which is only allowed if the data are derived from the same experiments, in which case should be clarified in all respective figure legends: 1D and 3C, 3B and 5A, Figure 3B and Supp figure 5B, Figure 5B and Appendix Supp figure 5B.
- Our production/data editors have asked you to clarify several points in the figure legends:
 - o Please note that the supplementary figures 2, 3, 4 is mislabeled as figures 3, 4, 5 in the manuscript. This needs to be rectified.
 - o Please note that the exact p values are not provided in the legends of figures 2C, D; 4C, 5B, E, G; 6A, B, D; supplementary figure(s) 1B, 3, 4.
 - o Please indicate what */ **/ ***/ **** represents; if this represents p value(s), please indicate the statistical test used and where appropriate, specify the exact p value in the legend(s) of figure(s) 3E.
 - o Please indicate what */ **/ ***/ **** represents; if this represents p value(s), please specify the exact p value in the legend(s) of supplementary figure(s) 5C-E.
 - o Please note that information related to n is missing in the legend of figure 2B.
 - o Please note that the error bars are not defined in the legends of figures 2B, 3E.
- Papers published in EMBO Reports include a 'synopsis' and 'bullet points' to further enhance discoverability. Both are displayed on the html version of the paper and are freely accessible to all readers. The synopsis includes a short standfirst summarizing the study in 1 or 2 sentences (max 35 words) that summarize the paper and are provided by the authors and streamlined by the handling editor. I would therefore ask you to include your synopsis blurb and 3-5 bullet points listing the key experimental findings.
- In addition, please provide an image for the synopsis. This image should provide a rapid overview of the question addressed in the study but still needs to be kept fairly modest since the image size cannot exceed 550 (width) x 300-600 (height) pixels.

Thank you again for giving us to consider your manuscript for EMBO Reports, I look forward to your minor revision.

Kind regards,

Deniz Senyilmaz Tiebe

--

Deniz Senyilmaz Tiebe, PhD
Senior Scientific Editor
EMBO Reports

Referee #3:

Authors have address my previous comment satisfactory.
Some errors in the revised edition:

Line 59: sentence too long. Advise splitting after 'respiratory chain' to improve clarity.

line 288: 'uniform expression of IF1 and sft1 under lactate and glycerol conditions'. I see this for SFT1/ β but not IF1/ β .

Figure 1 legend (B) there are four colours but legend only describe two.

Figure 3 E-G, Legends do not correspond to figures.

Figure 4 B and C, reduce complexity of legend. Would be best to say 'Growth of WT (Black circles and crosses) and inh1 Δ stf1 Δ (red circles and crosses)...' as you mention later on in legended that crosses are CCCP supplemented and circles no CCCP.

Supplementary figure 1, no densitometric analysis for atp18 Δ .

Supplementary figure 5D and 5E, legend says cdc19 Δ , figure says pyk1 Δ .

line 794 - space issue.

Referee #3:

Authors have address my previous comment satisfactory.
Some errors in the revised edition:

Line 59: sentence too long. Advise splitting after 'respiratory chain' to improve clarity.

line 288: 'uniform expression of IF1 and sft1 under lactate and glycerol conditions'. I see this for SFT1/ β but not IF1/ β .

Figure 1 legend (B) there are four colours but legend only describe two.

Figure 3 E-G, Legends do not correspond to figures.

Figure 4 B and C, reduce complexity of legend. Would be best to say 'Growth of WT (Black circles and crosses) and inh1 Δ stf1 Δ (red circles and crosses)...' as you mention later on in legended that crosses are CCCP supplemented and circles no CCCP.

Supplementary figure 1, no densitometric analysis for atp18 Δ .

Supplementary figure 5D and 5E, legend says cdc19 Δ , figure says pyk1 Δ .

line 794 - space issue.

Rev_Com_number: RC-2024-02639

New_manu_number: EMBOR-2024-60985V1-T

Corr_author: Mourier

Title: Novel If1 mechanism preventing ATP hydrolysis by the ATP synthase subcomplex in *S. cerevisiae*

université
de
BORDEAUX

Arnaud Mourier, Ph.D.

Group Leader BioDynaMit

Institut de Biochimie et de Génétique Cellulaires

1, rue Camille Saint-Saëns

33077 BORDEAUX cedex

+33556999040

arnaud.mourier@ibgc.cnrs.fr

New_manu_number: EMBOR-2024-60985V1-T

Corr_author: Mourier

Title: Novel If1 mechanism preventing ATP hydrolysis by the ATP synthase subcomplex in *S. cerevisiae*

Dear Dr Deniz Senyilmaz Tiebe,

We thank you for providing us with expert reviewers and we appreciate their very positive and constructive comments. We have carefully considered each of the reviewer 3 suggestions and have revised the manuscript accordingly. We have also formatted and corrected the manuscript according to your suggestions to comply to the EMBO Reports format.

- Please address the minor concerns of referee #3.
- Line 59: sentence too long. Advise splitting after 'respiratory chain' to improve clarity.
- line 288: 'uniform expression of IF1 and sft1 under lactate and glycerol conditions'. I see this for SFT1/β but not IF1/β.
- Figure 1 legend (B) there are four colours but legend only describe two.
- Figure 3 E-G, Legends do not correspond to figures.
- Figure 4 B and C, reduce complexity of legend. Would be best to say 'Growth of WT (Black circles and crosses) and inh1Δ stf1Δ (red circles and crosses)...' as you mention later on in legend that crosses are CCCP supplemented and circles no CCCP.
- Supplementary figure 1, no densitometric analysis for atp18Δ.
- Supplementary figure 5D and 5E, legend says cdc19Δ, figure says pyk1Δ.
- line 794 - space issue.

We would like to thank the reviewers for their time and effort in reviewing and improving the quality and clarity of our manuscript. We agree with all the suggested corrections, and have highlighted these changes in red in the manuscript.

- Please provide 3-5 keywords for your study. These will be visible in the html version of the paper and on PubMed and will help increase the discoverability of your work. *Done*
- As per our guidelines, please add a 'Data Availability Section', where datasets and computer code that were generated in the reported study should be listed in a structured manner and placed after the Methods section. If your study does not include datasets, please insert the following statement: This study includes no data deposited in external repositories (please see <https://www.embopress.org/page/journal/14693178/authorguide#dataavailability> for further information). *Done*
- Please add a Disclosure Statement & Competing Interests section to the manuscript (<https://www.embopress.org/page/journal/14693178/authorguide#conflictsofinterest>)
- Please remove the Author Contributions section from the manuscript text. *Done*
- As per our format requirements, in the reference list, citations should be listed in alphabetical order and then chronologically, with the authors' surnames and initials inverted; where there are more than 10 authors on a paper, 10 will be listed, followed by 'et al.'. Please see <https://www.embopress.org/page/journal/14693178/authorguide#referencesformat> *Done*
- Please fill out and include an author checklist as listed in our online guidelines (<https://www.embopress.org/page/journal/14693178/authorguide>) *Done*
- We note that the funding information regarding University of Bordeaux is currently missing from the manuscript tracking system. *Done*
- We note the following regarding the figure files:

- o The main and supplementary figures provided in one PDF. Main figures need to be uploaded as separate production quality Figure files. *Done*
- o The correct nomenclature for supplemental figures is Expanded View figures and their callouts are Figure EV1 etc. EV Figures also need to be provided as separate production quality Figure files. *Done*
- o Suppl. figure 2 and its legend are currently missing, and it is not called out in the text.
- o The figure PDF is not properly displayed in the Merged PDF - panel 1F is missing, Suppl. figure 2 is missing, panels 3EFG are missing, etc. *Done*
- All research articles submitted as revised versions must include a structured methods section that includes a Reagents and Tools Table followed by a Methods and Protocols section. This file should be submitted separately. Please see <https://www.embopress.org/page/journal/14693178/authorguide#structuredmethods> for further information. *Done*
- Please submit source data along with the revised manuscript as requested by our Source Data Coordinator Dr. Hannah Sonntag per email on 20.01.2025. *Done*
- The manuscript sections should be in the following order: Title page - Abstract & Keywords - Introduction - Results - Discussion - Methods - Data Availability - Acknowledgments - Disclosure Statement & Competing Interests - References - Figure Legends - (Main Tables with legends if applicable) - Expanded View Figure Legends. *Done*
- "Mat & meth" should be renamed as "Methods". *Done*
- During our routine figure checks, we note the following potential panel re-uses, which is only allowed if the data are derived from the same experiments, in which case should be clarified in all respective figure legends: 1D and 3C, 3B and 5A, Figure 3B and Supp figure 5B, Figure 5B and Appendix Supp figure 5B. *We have clarified this point in figure legend (1D & 3C) and we have clarified the re-use of the Coomassie staining validating the equal levels of If1 and Stf1 in the standard used in 3B, 5A, EVIF in the Methods section.*
- Our production/data editors have asked you to clarify several points in the figure legends:
 - o Please note that the supplementary figures 2, 3, 4 is mislabeled as figures 3, 4, 5 in the manuscript. This needs to be rectified. *Done*
 - o Please note that the exact p values are not provided in the legends of figures 2C, D; 4C, 5B, E, G; 6A, B, D; supplementary figure(s) 1B, 3, 4. *Done*
 - o Please indicate what */ **/ ***/ **** represents; if this represents p value(s), please indicate the statistical test used and where appropriate, specify the exact p value in the legend(s) of figure(s) 3E. *Done*
 - o Please indicate what */ **/ ***/ **** represents; if this represents p value(s), please specify the exact p value in the legend(s) of supplementary figure(s) 5C-E. *Done*
 - o Please note that information related to n is missing in the legend of figure 2B. *Done*
 - o Please note that the error bars are not defined in the legends of figures 2B, 3E. *Done*
- Papers published in EMBO Reports include a 'synopsis' and 'bullet points' to further enhance discoverability. Both are displayed on the html version of the paper and are freely accessible to all readers. The synopsis includes a short standfirst summarizing the study in 1 or 2 sentences (max 35 words) that summarize the paper and are provided by the authors and streamlined by the handling editor. I would therefore ask you to include your synopsis blurb and 3-5 bullet points listing the key experimental findings. *Done*
- In addition, please provide an image for the synopsis. This image should provide a rapid overview of the question addressed in the study but still needs to be kept fairly modest since the image size cannot exceed 550 (width) x 300-600 (height) pixels. *Done*

We thank you in advance for your time and consideration.

Sincerely yours,

Arnaud MOURIER,
Ph.D., corresponding author

Dr. Arnaud Mourier
IBGC (CNRS-UMR5095)
BioDynaMit
1, rue Camille Saint-Saëns
Bordeaux 33000
France

Dear Arnaud,

Thank you for submitting your revised manuscript. I have now looked at everything and all is fine. Therefore, I am very pleased to accept your manuscript for publication in EMBO Reports.

Congratulations on a nice work!

Kind regards,

Deniz
--
Deniz Senyilmaz Tiebe, PhD
Senior Scientific Editor
EMBO Reports

Rev_Com_number: RC-2024-02639
New_manu_number: EMBOR-2024-60985V2
Corr_author: Mourier
Title: Novel If1 mechanism preventing ATP hydrolysis by the ATP synthase subcomplex in *S. cerevisiae*